# The Complex World of Emaraviruses—Challenges, Insights, and Prospects

**Marius Rehanek** [1], **David G. Karlin** [1,2], **Martina Bandte** [1], **Rim Al Kubrusli** [1],
**Shaheen Nourinejhad Zarghani** [1], **Thierry Candresse** [3], **Carmen Büttner** [1] and **Susanne von Bargen** [1,*]

1 Division Phytomedicine, Thaer-Institute of Agricultural and Horticultural Sciences, Humboldt-Universität zu Berlin, Lentzeallee 55/57, D-14195 Berlin, Germany
2 Independent Researcher, 13000 Marseille, France
3 UMR BFP, INRAE, University of Bordeaux, 33140 Villenave d'Ornon, France
* Correspondence: susanne.von.bargen@agrar.hu-berlin.de; Tel.: +49-30-2093-46447

**Abstract:** *Emaravirus* (Order *Bunyavirales*; Family *Fimoviridae*) is a genus comprising over 20 emerging plant viruses with a worldwide distribution and economic impact. Emaraviruses infect a variety of host plants and have especially become prevalent in important long-living woody plants. These viruses are enveloped, with a segmented, single-stranded, negative-sense RNA genome and are transmitted by eriophyid mites or mechanical transmission. Emaraviruses have four core genome segments encoding an RNA-dependent RNA polymerase, a glycoprotein precursor, a nucleocapsid protein, and a movement protein. They also have additional genome segments, whose number varies widely. We report here that the proteins encoded by these segments form three main homology groups: a homolog of the sadwavirus Glu2 Pro glutamic protease; a protein involved in pathogenicity, which we named "ABC"; and a protein of unknown function, which we named "P55". The distribution of these proteins parallels the emaravirus phylogeny and suggests, with other analyses, that emaraviruses should be split into at least two genera. Reliable diagnosis systems are urgently needed to detect emaraviruses, assess their economic and ecological importance, and take appropriate measures to prevent their spread (such as routine testing, hygiene measures, and control of mite vectors). Additional research needs include understanding the function of emaravirus proteins, breeding resistant plants, and clarifying transmission modes.

**Keywords:** diagnosis; distribution; emaraviruses; forest trees; *Fimoviridae*; genome organization; phylogenetic relations; protein domains; symptomatology; transmission

## 1. Introduction

Emaraviruses are an emerging group of plant-infecting, segmented negative-sense RNA viruses with enveloped particles. They are currently classified as a single genus in the new *Fimoviridae* family (order *Bunyavirales*) [1] and were discovered relatively late by plant virologists. Several virus-like diseases that resisted efforts for a long time to identify the causal agents have now been linked to emaraviruses. For example, viruses causing the rosette disease of roses, the mosaic disease of figs, the sterility mosaic disease of pigeonpea, the ringspot diseases of rowan and oak, or the mosaic disease of aspen trees, eluded the efforts of scientists for decades due to a lack of suitable methods for identification and characterization of such viruses.

Virus-like symptoms such as chlorotic ringspots and mottle on diseased *Sorbus* spp. were first described by Baur [2] and later Kegler [3]. Büttner and Führling [4] reported on the occurrence and distribution of diseased oak trees in Germany, and Ebrahim-Nesbat [5] showed the first electron microscopic images of virus-like particles associated with ringfleck mosaic of mountain ash. A decade later, the identification of a double-stranded RNA pattern and partial sequence data were a cornerstone for the association of the ringspot disease

of mountain ash with an unknown virus [6]. The virus, later characterized as European mountain ash ringspot-associated virus (EMARaV) became the first member of a novel virus genus [7,8]. The first description and genetic characterization of EMARaV initiated the identification of similar agents associated with well-known diseases of fig [9], maize [10], and pigeonpea [11,12]. Following the discovery of further related viruses, the International Committee on Taxonomy of Viruses (ICTV) established the newly unassigned genus *Emaravirus* in 2012 with EMARaV, fig mosaic virus (FMV), High Plains wheat mosaic virus (HPWMoV, syn. maize red stripe virus (MRSV), syn. High plains virus (HPV)), pigeonpea sterility mosaic virus (PPSMV), raspberry leaf blotch virus (RLBV), and rose rosette virus (RRV) [8,13]. In 2018, the genus *Emaravirus* was assigned to the novel family *Fimoviridae*, in the *Bunyavirales* order [1]. In the following years, the number of emaraviruses has further increased. Especially in broad-leaved trees, new species of emaraviruses have exclusively been described, including EMARaV in *Sorbus intermedia* [14], *Karpatiosorbus* × *hybrida* [15], and *Amelanchier* sp. [16], aspen mosaic-associated virus (AsMaV) in *Populus tremula* [17], common oak ringspot-associated virus (CORaV) in *Quercus robur* [18,19], maple mottle-associated virus (MaMaV) in *Acer pseudoplatanus* [20], and ash shoestring-associated virus (ASaV) in *Fraxinus* spp. [21]. Emaraviruses have thereby become the most prevalent group of viruses in long-living plants, causing diseases in key species of the temperate and boreal forests.

To date, 24 species are included in the genus *Emaravirus* (Table 1). Emaraviruses are related to tospoviruses and peribunyaviruses within the *Bunyavirales*, in that they share (i) enveloped virions; (ii) segmented genomes; (iii) high sequence similarity in orthologous proteins for the RNA-dependent RNA polymerase (RdRP, RNA 1), the glycoprotein precursor (GPP, RNA 2), and the nucleocapsid protein (NC, RNA 3); (iv) conserved motifs in RdRP amino acid sequence; and (v) conserved terminal ends of each RNA segment that are nearly complementary to each other [1]. Using electron microscopy, emaraviruses can be visualized in the cytoplasm of infected cells as spherical enveloped particles of 80–100 nm in diameter (Figure 1). Many of them are transmitted by eriophyid gall mites [8].

**Table 1.** Overview of identified emaraviruses. Species recognized by the ICTV (italics) and putative emaraviruses are indicated chronologically, according to the time of their discovery.

| Species | Common Name | Disease/ Symptoms | Host(s) (Botanical Family) | Geographic Occurence | Vector/ Transmission | Genomic Segments | Key Reference(s) |
|---|---|---|---|---|---|---|---|
| Established Species (ICTV) | | | | | | | |
| *Emaravirus sorbi* | European mountain ash ringspot-associated virus (EMARaV) | mosaic and ringspot disease | rowan, serviceberry (*Sorbus* spp., *Amelanchier* sp.) and other *Rosaceae* species | North and Central Europe | *Phytoptus pyri*, graft transmission and mechanical inoculation to *Sorbus aucuparia* L. | 6 | [7,14,16,22,23] |
| *Emaravirus fici* | fig mosaic virus (FMV) | fig mosaic disease (FMD) | fig, cyclamen (*Ficus carica, F. pseudocarica, Cyclamen persicum* Mill.) (*Moraceae, Primulaceae*) | North America, Europe, North Africa, Middle East, Asia, New Zealand | *Aceria ficus*, vegetative propagation and grafting | 6 | [9,24,25] |
| *Emaravirus rosae* | rose rosette virus (RRV) | rose rosette disease (RRD) | rose (*Rosa* spp.) (*Rosaceae*) | USA, Canada, India | *Phyllocoptes fructiphilus*, grafting | 6 | [26,27] |
| *Emaravirus idaeobati* | raspberry leaf blotch virus (RLBV) | raspberry leaf blotch disorder (RLBD) | raspberry (*Rubus* spp.) (*Rosaceae*) | Scandinavia, UK, Balkans | *Phyllocoptes gracilis*, mechanical transmission to experimental hosts | 8 | [28,29] |

**Table 1.** *Cont.*

| Species | Common Name | Disease/ Symptoms | Host(s) (Botanical Family) | Geographic Occurence | Vector/ Transmission | Genomic Segments | Key Reference(s) |
|---|---|---|---|---|---|---|---|
| *Emaravirus cajani* | pigeonpea sterility mosaic virus 1 (PPSMV-1) | sterility mosaic disease (SMD) | pigeonpea (*Cajanus* spp.) (*Fabaceae*) | South Asia | *Aceria cajani*, grafting, transmission to experimental hosts by sap inoculation and viruliferous eriophyid mites | 6 | [12,30,31] |
| *Emaravirus toordali* | pigeonpea sterility mosaic virus 2 (PPSMV-2) | sterility mosaic disease (SMD) | pigeonpea (*Cajanus* spp.) (*Fabaceae*) | South Asia | *Aceria cajani*, grafting, transmission to experimental hosts by sap inoculation and viruliferous eriophyid mites | 6 | [32] |
| *Emaravirus tritici* | High Plains wheat mosaic virus (HPWMoV) | High Plains disease (HPD) | Sweet grasses (wheat, maize, barley, oat, rye, cheat, green and yellow foxtail) (*Poaceae*) | Central and Western USA, Canada, Argentina, Australia, Ukraine | *Aceria tosichella* | 8 | [10,33] |
| *Emaravirus cercidis* | redbud yellow ringspot-associated virus (RYRaV) | redbud yellow ringspot (RYRS) | redbud (*Cercis* spp.) (*Fabaceae*) | USA | graft transmissible to Cercis 'Canadensis' and several legume species | 5 | [34] |
| *Emaravirus actinidiae* | Actinidia chlorotic ringspot-associated virus (AcCRaV) | chlorotic ringspot, mottle and vein yellowing of leaves | kiwifruit (*Actinidia* spp.) (*Actinidiaceae*) | China | mechanical transmission to *N. benthamiana* | 5 | [35] |
| *Emaravirus rubi* | blackberry leaf mottle-associated virus (BLMaV) | blackberry yellow vein disease (BYVD) | blackberry (*Rubus* spp.) (*Rosaceae*) | USA | undescribed gall mite species of *Eriophyidae* | 5 | [36] |
| *Emaravirus pistaciae* | Pistacia virus B (PiVB) | unknown | pistachio (*Pistacia* spp.) (*Anacardiaceae*) | Turkey | / [1] | 7 | [37] |
| *Emaravirus parkinsoniae* | palo verde broom virus (PVBV) | witches' broom disease | blue palo verde (*Parkinsonia florida*) (*Fabaceae*) | USA, Mexico | *Aculus cercidi*? | 4 | [38] |
| *Emaravirus kiwi* | Actinidia virus 2 (AcV-2) | leaf mottle, chlorotic spots and/or leaf mosaic symptoms | kiwifruit (*Actinia* spp.) (*Actinidiaceae*) | China | / | 6 | [39] |
| *Emaravirus ziziphi* | jujube yellow mottle-associated virus (JYMaV) syn. Chinese date mosaic-associated virus (CDMaV) | jujube yellow mottle disease (JYMD) | jujube (*Ziziphus jujuba*) (*Rhamnaceae*) | China | *Epitrimerus zizyphagus*? | 6 | [40,41] |
| *Emaravirus cordylinae* | ti ringspot-associated virus (TiRSaV) | ti ringspot | ti (*Cordyline fructicosa*) (*Asparagaceae*) | Hawaii | undescribed species of eriophyid mites, mechanical transmission to experimental hosts | 5 | [42] |

**Table 1.** *Cont.*

| Species | Common Name | Disease/ Symptoms | Host(s) (Botanical Family) | Geographic Occurence | Vector/ Transmission | Genomic Segments | Key Reference(s) |
|---|---|---|---|---|---|---|---|
| *Emaravirus camelliae* | Camellia japonica-associated virus 1 (CjaV-1) syn. Camellia chlorotic ringspot virus 1 (CaCRSV-1) | chlorotic ringspots, color-breaking | common camellia (*Camellia japonica*) (*Theaceae*) | Italy, China | / | 9 | [43–45] |
| *Emaravirus verbanni* | Camellia japonica-associated virus 2 (CjaV-2) syn. Camellia chlorotic ringspot virus 2 (CaCRSV-2) | chlorotic ringspots | common camellia (*Camellia japonica*) (*Theaceae*) | Italy, China | / | 4 | [43] |
| *Emaravirus perillae* | Perilla mosaic virus (PerMV) | mosaic disease | shiso (*Perilla* spp.) (*Lamiaceae*) | Japan | perilla rust mite (*Shevtchenkella* spp.) mechanical transmission to *N. benthamiana* | 10 | [46] |
| *Emaravirus populi* | aspen mosaic-associated virus (AsMaV) | mosaic disease | Euroasien aspen (*Populus tremula*) (*Saliaceae*) | Fennoscandinavia | mite vector species unknown, graft-transmissible | 5 | [17] |
| *Emaravirus syringae* | lilac chlorotic ringspot-associated virus (LiCRaV) | leaf chlorotic ringspots and mottling | common lilac (*Syringa vulgaris* L.) (*Oleaceae*) | China | mechanical transmission to *N. benthamiana* | 5 | [47] |
| *Emaravirus pyri* | pear chlorotic leaf spot-associated virus (PCLSaV) | chlorotic leaf spot disease | sandy pear (*Pyrus pyrifolia*) (*Rosaceae*) | Central and Southern China | / | 5 | [48] |
| *Emaravirus quercus* | common oak ringspot-associated virus (CORaV) | ringspot disease | common oak (*Quercus robur* L.) (*Fabaceae*) | North and Central Europe | graft-transmissible | 5 | [18,19,22] |
| *Emaravirus aceris* | maple mottle-associated virus (MaMaV) | mottle, mosaic, chlorotic ringspots | sycamore maple (*Acer pseudoplatanus*) (*Sapindaceae*) | Germany | graft-transmissible | 6 | [20,49] |
| *Emaravirus chrysantemi* | chrysanthemum mosaic-associated virus (ChMaV) | 'Mon-mon' disease (chlorotic ringspots, mosaic) | Chrysanthemum (*Asteraceae*) | Japan | *Paraphytoptus kikus*? | 7 | [50] |
| putative emaravirus based on genomic, biological, and phylogenetic properties | | | | | | | |
| | alfalfa ringspot-associated virus (ARaV) | ringspot, yellow mosaic | alfalfa (*Medicago sativa* L.) (*Fabaceae*) | Australia, China | / | 4 | [51,52] |
| | Vitis emaravirus (VEV) syn. grapevine emaravirus A (GEVA) | chlorotic mottling | grapevine (*Vitis coignetiae*, *Vitis vinifera* L.) (*Vitaceae*) | Japan, China | graft-transmissible | 5 | [53,54] |
| | ash shoestring-associated virus (ASaV) syn. pea associated emaravirus (PaEV) | shoestring, leaf curling, mottle, chlorotic spots and ringspots | ash, pea (*Fraxinus* spp., *Pisum sativum* L.) (*Oleaceae*, *Fabaceae*) | Germany, Switzerland, Sweden, Italy | *Aceria fraxinivora*? | 5 | [21,55] |

**Table 1.** *Cont.*

| Species | Common Name | Disease/ Symptoms | Host(s) (Botanical Family) | Geographic Occurence | Vector/ Transmission | Genomic Segments | Key Reference(s) |
|---|---|---|---|---|---|---|---|
| | karaka Okahu purepure virus (KOPV) | chlorotic pale spots | karaka (*Corynocarpus laevigatus*) (*Corynocarpaceae*) | New Zealand | *Aculus corynocarpi*? | 5 | [56] |
| | Japanese star anise ringspot-associated virus (JSARaV) | ringspot disease | Japanese star anise (*Illicium anisatum* L.) (*Schisandraceae*) | Japan | eriophyid mites of the family *Diptilomiopidae* | 5 | [57] |
| | Arceuthobium sichuanense-associated virus 1 (ArSaV1) | unknown | spruce dwarf mistletoe (*Arceuthobium sichuanense*) (*Viscaceae*) | China | / | 5 | [58] |
| | Artemisia fimovirus 1 (ArtV1) | unknown | Chinese mugwort (*Artemisia verlotiorum*) (*Asteraceae*) | Slovenia | / | 5 | [59] |
| | Ailanthus crinkle leaf-associated emaravirus (ACrLaV) | severe crinkle | tree of heaven (*Ailanthus altissima*) (*Simaroubaceae*) | China | / | 4 | [60] |
| | Pueraria lobata-associated emaravirus (PloAEV) | yellow spots, mosaic, mottling | kudzu (*Pueraria lobata*) (*Fabaceae*) | China | mechanical transmission to *N. benthamiana* | 5 | [61] |
| Orphans * | | | | | | | |
| | Woolly burdock yellow vein virus (WBYVV) | yellow vein and leaf mosaic symptoms | wooly burdock (*Arctium tomentosum*) (*Asteraceae*) | Finland | / | 2 | [62] |
| | Yunnan emara-like virus | / | / | China | / | 7 | [63] |
| | Illicium anisatum ringspot-associated virus | / | Japanese star anise (*Illicium anisatum* L.) (*Schisandraceae*) | / | / | 1 | Shimomoto and Kubota, unpublished |
| / | / | / | *Solanum lichtensteinii, S. mauritianum* (*Solanaceae*) | South Africa | / | 1 | [64] |

* Based on experimental evidence, a clear assignment to the genus *Emaravirus* has yet to be demonstrated. For more details see Section 8.2. *[1] no information available.

Emaravirus infections often induce macroscopically visible leaf symptoms as described in Section 2. Such symptoms can be observed on many different host plants that are susceptible to infection by various emaraviruses. The host range and mode of emaravirus transmission is summarized in Section 3. In Section 4, we report the geographic distribution of emaraviruses which encompasses all continents.

The genomes of emaraviruses differ in size due to their variable number of genome segments. It should be noted, however, that this perhaps reflects an incomplete characterization of some members. However, as indicated above, a core genome of four RNAs encoding the polymerase (RNA 1), the glycoprotein precursor (RNA 2), the nucleocapsid (RNA 3), and a movement protein (RNA 4) has been confirmed for each emaravirus described to date. In Section 6, we summarize the genome organization of emaraviruses including all accessory segments, the taxonomic considerations of which are discussed in Section 7.

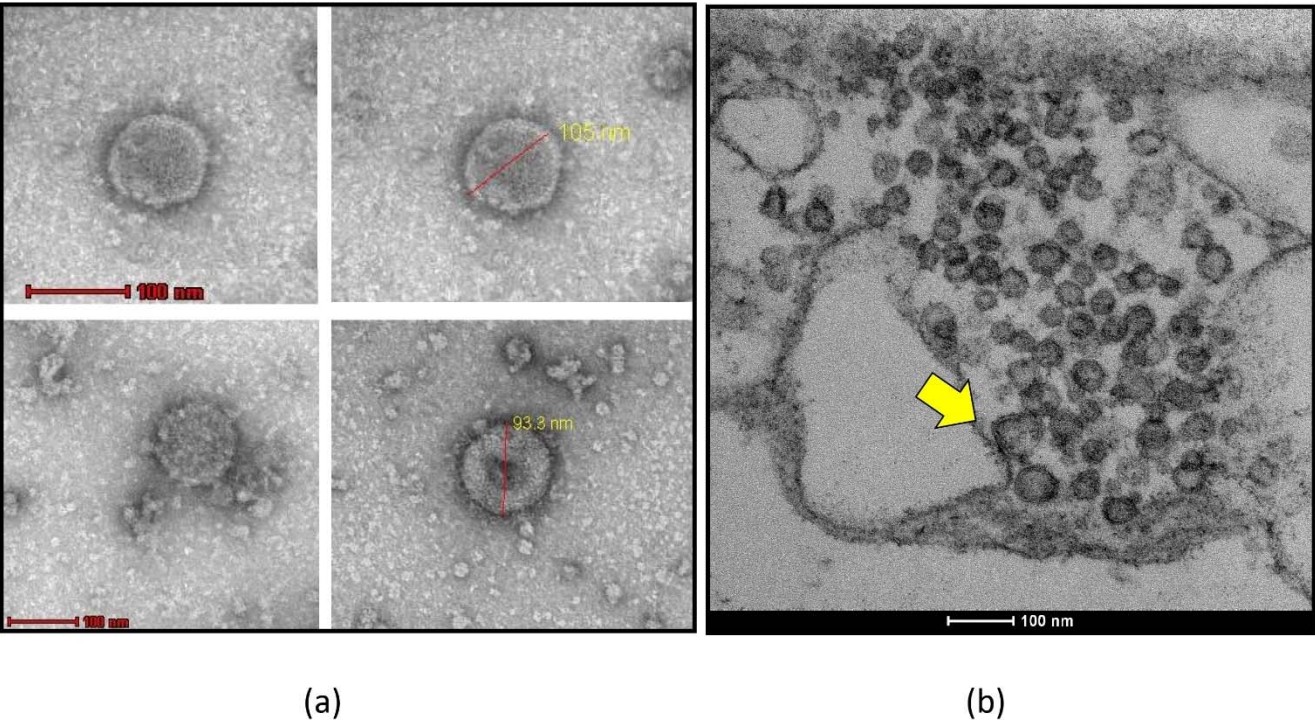

(a)                                                                (b)

**Figure 1.** Transmission electron microscopic (TEM) visualization of negatively stained emaravirus particles. (**a**) Spherical particles of approx. 100 nm found in plant homogenates of *Nicotiana tabacum* 10 days after inoculation with ash shoestring associated-virus (ASaV) following negative staining with 2% ammonium molybdate. (**b**) Ultrathin section of mesophyll cells of symptomatic, ASaV-infected common ash (*Fraxinus excelsior*) showing double membrane-bound bodies (DMB, yellow arrow) typically found in emaravirus-infected tissues [15].

A description of recently detected viruses (Table 1) that are potential genus members is given in Section 8. The majority have been detected using high-throughput sequencing technologies. For diagnosis, nucleic acid-based and serological means are available, which are described in Section 9.

The complex world of emaraviruses has just begun to unravel. Gaining more knowledge on their biology and epidemiology is needed to efficiently handle the challenges we face concerning this emerging group of plant pathogenic viruses.

## 2. Symptomatology

Symptoms on host plants affected by emaraviruses differ tremendously depending on the emaravirus-host combination. The type and severity of symptoms depend on the virulence of the infecting virus strain/isolate/variant/serotype as well as the susceptibility or tolerance of the host plant species/varieties/cultivars.

The most common symptoms are mosaic, mottling, blotching, yellowing, and vein clearing. Chlorotic ringspots were also often detected on leaves of diseased plants (Figure 2). In many cases, emaraviruses are associated with typical leaf symptoms. The most prominent symptoms observed on affected host plants were frequently used to name the corresponding virus species.

Virus-like diseases that are now associated with emaraviruses have been known for a long-time. Mitra (1931) first observed characteristic symptoms of the sterility mosaic disease (SMD) of pigeonpea [65], 70 years before they were demonstrated to be caused by virus infection with pigeonpea sterility mosaic virus 1 (PPSMV-1) and pigeonpea sterility mosaic virus 2 (PPSMV-2) [30,32]. Reports of diseased roses in North America and Canada also reach back to the 1940s and fig mosaic disease was already described in the 1930s [66].

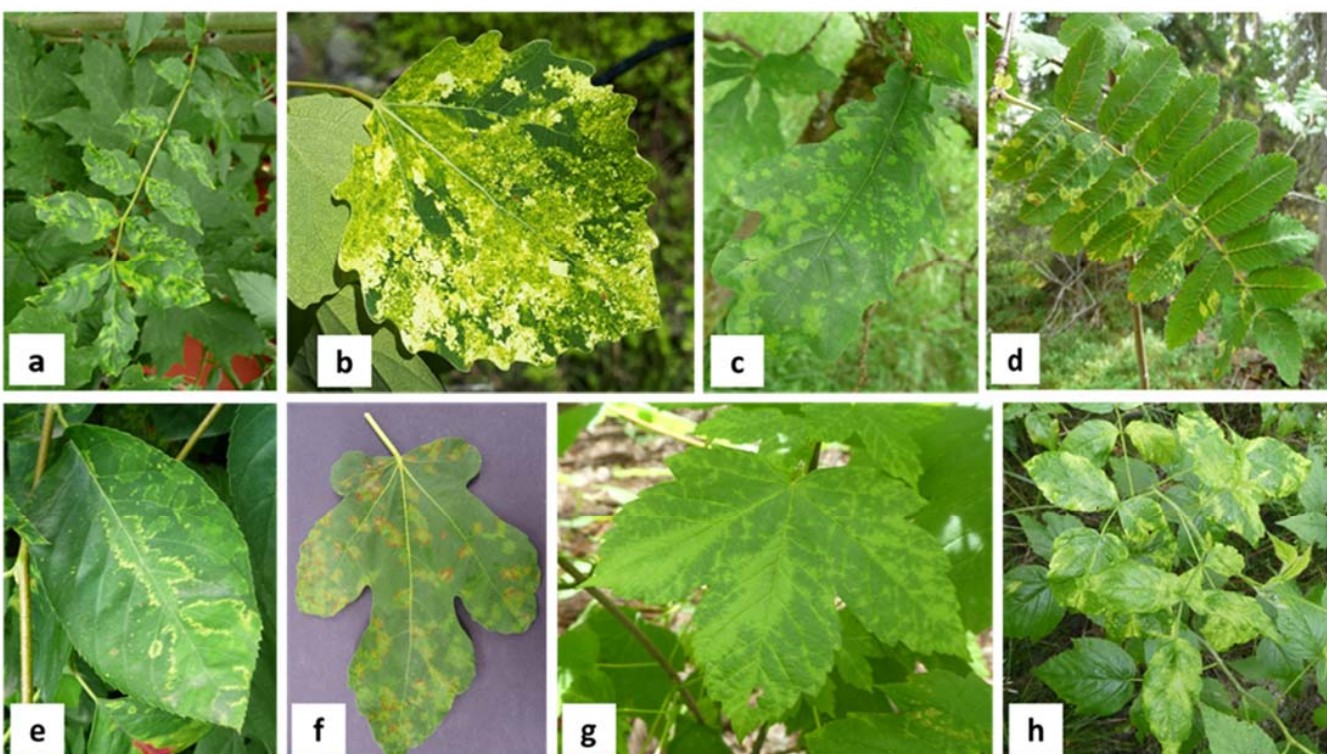

**Figure 2.** Typical symptoms on leaves of emaravirus-infected woody plant species detected in forests or urban green. (**a**) Chlorotic spots, leaf curling, and beginning of shoestring symptoms caused by ash shoestring-associated virus (ASaV) in common ash (*Fraxinus excelsior*), (**b**) mosaic caused by aspen mosaic-associated virus (AsMaV) in Eurasian aspen (*Populus tremula*), (**c**) chlorotic ringspots caused by common oak ringspot-associated virus (CORaV) in common oak (*Quercus robur*), (**d**) chlorotic ringspots caused by European mountain ash ringspot-associated virus (EMARaV) in rowan (*Sorbus aucuparia*), (**e**) chlorotic ringspots and line pattern caused by EMARaV in serviceberry (*Amelanchier lamarckii*), (**f**) chlorotic ringspots and mosaic caused by fig mosaic virus (FMV) in fig (*Ficus carica*), (**g**) mottle and vein yellowing caused by maple mottle-associated virus (MaMaV) in sycamore maple (*Acer pseudoplatanus*), (**h**) yellow blotches caused by raspberry leaf blotch virus (RLBV) in raspberry (*Rubus idaeus*).

Symptom expression is influenced by many factors including the virus and host genotype as well as the plant age at the time of virus infection expression [67]. For example, SMD which is caused by pigeonpea infecting emaraviruses PPSMV-1 and PPSMV-2 is divided into three types. Diseased plants either develop severe mosaic and sterility, mild mosaic and partial sterility, or chlorotic ringspot symptoms without sterility [31].

However, some studies have also revealed that emaraviruses can infect host plants in a latent way with plants remaining symptomless during early infection, enabling the virus to spread inconspicuously. Latent infection with emaraviruses have been reported, for instance in RRV-infected roses [68], EMARaV-infected rowan [69], pistachios infected by Pistachia virus B (PiVB) [37], pear chlorotic leaf spot-associated virus (PCLSaV)-infected pear [48], and ASaV-infected ash [21].

For emaraviruses infecting long-living woody hosts, visual inspection and sampling of big tree crowns is difficult and often requires technical support. Together with the irregular virus distribution in trees, this might lead to an overlook of symptoms. Therefore, a deeper look in the crowns is necessary to unravel data on virus distribution and virus infection for old trees. In extreme cases, perennial plants affected by certain emaraviruses have been reported to show decline, which can result in the death of trees or shrubs. For instance, RRV-infected roses often display a general decline leading to plant death [26], and blackberry

leaf mottle-associated virus (BLMaV) was found in a significant number of blackberry yellow vein-diseased blackberry plants, resulting in a decline of infected blackberries [36].

Identifying a clear association between diseases and emaraviruses remained challenging for a long time. This is because emaravirus-infected host genera are known to be associated with numerous viruses. *Rubus* plants are susceptible to infection by several viruses and virus-like agents [70]. Besides RRV, roses can be affected by viruses belonging to seven other genera [71]. In figs, 15 viruses have been identified, belonging to several genera [72–77]. The recently detected Pueraria lobata-associated emaravirus (PloAEV) was shown to occur only in mixed infection with two different viruses raising questions of mutual relationships between them [61].

The complexity of viruses in different combinations can induce a variety of symptoms with synergistic or antagonistic effects in their hosts, thereby impeding the determination of which symptom is caused by which virus. Additionally, the identification of emaraviruses came relatively late in the history of plant virology as the physicochemical properties of their enveloped particles complicated their purification and therefore slowed research based on classical virology approaches.

An emaravirus infection can also affect fruit quality, as described for example in jujube yellow mottle-associated virus (JYMaV)-infected jujube, whereby fruits with distortion, malformation, discoloration, and necrotic areas around the calix were observed [40]. Similarly, reduced fruit quality was also reported for *Rubus* spp. suffering with blackberry yellow vein disease (BYVD) attributed to BLMaV infection [36,70].

## 3. Transmission and Host Range

Emaraviruses can infect a variety of host plants but individual members seem to have limited natural host ranges, often being reported to naturally infect a single host species. This might reflect the transmission biology of these viruses and the very narrow host range of their mite vectors.

### 3.1. Transmission

Emaraviruses are transmitted by eriophyid mites, also referred to as gall mites, which are small plant-sucking arthropods of 140–170 μm in size [78]. Mites of the following genera have been identified as vectors of emaraviruses: *Phytoptus* [79], *Aceria* [80–83], *Phyllocoptes* [84], and *Shevtchenkella* [46]. Additionally, an eriophyid mite of the family *Diptilomiopidae* was demonstrated to transmit a novel emaravirus to star anise [57]. New, yet unclassified eriophyid mite species are suspected to be vectors for other putative emaraviruses (refs. [21,36,38,41,42,50], see Table 1). However, vector transmission has so far not been demonstrated for all emaraviruses.

Eriophyid mites can often be found on leaf petioles and axillary buds and overwinter on their host plant (Figure 3). They can be transported by insects during pollination, dispersed by wind, or by contact with clothing, thus contributing to natural virus dissemination [85].

Also, little is known about the underlying mechanism(s) of transmission. Eriophyid mite species transmitting PPSMV-1, PPSMV-2, and perilla mosaic virus (PerMV) [46,82] are known to acquire the virus rapidly, as acquisition times of 15 min and 30 min were sufficient for successful uptake and transmission to new host plants. In contrast, the nucleocapsid protein of EMARaV is not restricted to the mites mouth part of the vector *P. pyri* [79]. FMV and HPWMoW were reported to be transmitted in a persistent manner by *A. ficus* and *A. tosichella* [83,86]. It also remains unclear whether emaraviruses replicate within their vectors as reported for related tospoviruses [87].

Transmission through vegetative propagation practices such as grafting or cuttings has also been demonstrated for FMV [88], PPSMV-1 [31], EMARaV and CORaV [22], RRV [26], redbud yellow ringspot-associated virus (RYRaV) [34], and AsMaV [17]. Experimentally, emaraviruses can be mechanically transmitted; however, it is unclear if this significantly contributes to their spread in nature.

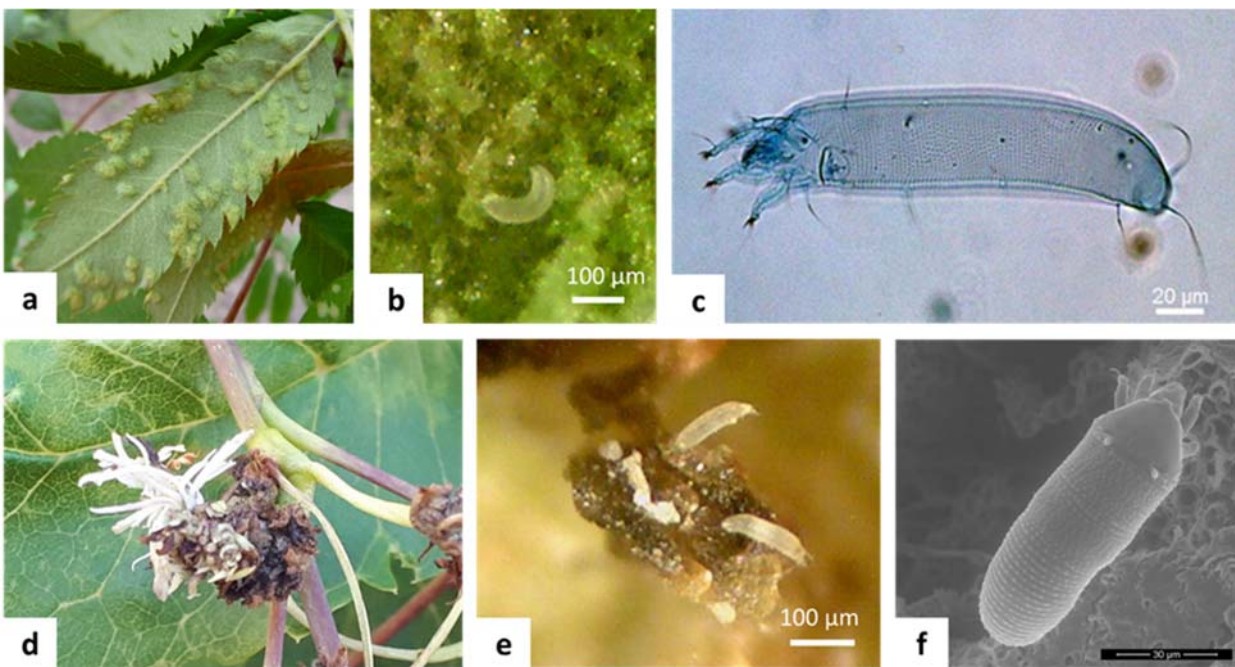

**Figure 3.** Gall mite symptoms and gall mite vectors associated with emaraviruses. (**a**) Galls induced by the pear leaf blister mite *Phytoptus pyri* (Pagenstecher) on the underside of an *S. aucuparia* leaf, (**b**) *P. pyri* in the erineum of the galls and (**c**) close up of the mite under a light microscope. (**d**) Cauliflower galls induced by the mite *Aceria fraxinivora* (Nalepa) in inflorescences of *Fraxinus ornus* and *F. excelsior*, (**e**) close up of *A. fraxinivora* mites inside the galls, and (**f**) detail of the mites as seen in electron microscopy, showing the characteristic prodorsal shield.

There is currently little information on further modes of emaravirus transmission, namely by seed or pollen or through water or soil. Gained experience on water and soil transmission of other viruses over many years lead us to the assumption that a transmission through water and soil should be taken into consideration [89]. Although RRV could be detected in rose pollen, this result has to be confirmed and the transmission through pollen evaluated in detail [90]. For CORaV, a screening of seedlings grown from acorns originating from virus-infected trees revealed the infection of a single oak seedling, indicating that emaraviruses might possibly be transmitted by seeds [91].

### 3.2. Host Range

Emaraviruses infect a variety of hosts including herbaceous annual and woody perennial plants, fruit and field crops as well as ornamental and wild plant species (ref. [92]; Table 1). Most natural host plants belong to the Eudicots, and only HPWMoV and ti ringspot-associated virus (TiRSaV) are known to infect monocots. With only few exceptions, emaraviruses seem to establish long-lasting infections in a restricted number of host species, mainly comprising of a single genus. However, transmission to experimental herbaceous plants including *Arabidopsis thaliana*, *Nicotiana* (*N.*) *benthamiana*, *N. tabacum*, *N. clevelandii*, *Phaseolus vulgaris*, *Chrozophora rottleri*, and various cucurbit species was demonstrated by mechanical means for pigeonpea-infecting emaraviruses [31], RLBV [28], AcCRaV [35], TiRSaV [42], PerMV [46], LiCRaV [47], RRV [93], and PloAEV [61]. In greenhouse experiments, several sweet grasses were reported to be infected with HPWMoV [94]. Besides rowans, EMARaV was reported in other *Sorbus* species, as well as in serviceberry (*Amelanchier* sp.) and several other *Rosaceae* species [16,23]. In addition to its primary natural host the common fig, FMV was reported in cyclamen [95] and wild fig (*F. pseudocarica*) [96]. FMV and ASaV are the only species known to date to be detected in hosts belonging to different plant families. Spontaneous host plants in marginal and uncultivated areas may serve as reservoirs, when the sensitive and elective host plant is temporarily unavailable. The

ability of emaraviruses to be transmitted to species other than their natural host and to have additional natural hosts emphasizes the importance of effective management strategies (see later).

## 4. Geographic Occurrence

Emaraviruses are distributed all over the world, with single species being mainly restricted to one or two continents. There are two emaraviruses that are reported to have a wider distribution. FMV has been detected worldwide in fig-growing areas [77] and HPWMoV has been identified in America, Europe, and Australia [10,97,98]. For the other species, there are three hotspots, discussed in the following, including North America, Europe, and Asia (Figure 4).

1. In North America, endemic emaraviruses have been reported in large areas of the US and Canada. HPWMoV was reported from several US states comprising the Great Plains region [10,99,100]. RRV, initially reported from the Eastern USA, has also been detected in 36 US states including middle and western states and in East Canada [101,102]. Outside North America, RRV has so far only been reported in India [103]. In the Eastern United States, BLMaV was detected [36]. RYRaV, palo verde broom virus (PVBV), and TiRSaV have been reported in spatially limited regions.

2. In Europe, RLBV has been reported from several northern countries [104], western countries [28], and the Balkan regions [105,106]. In recent years, a constantly growing number of emaraviruses affecting important deciduous tree species have been detected. Since its first description in German rowan trees in 2005 [6], EMARaV has been reported in several European countries including Austria, the Czech Republic, Sweden, Finland, Norway, and the United Kingdom [107]. Novel emaraviruses affecting oak, sycamore, ash, and aspen trees in forests and urban landscapes were detected in several Northern and Central European countries [17,19–21].

3. In Asia, particularly in the southern and eastern region, a variety of emaraviruses affecting important commercial fruits and ornamental crops have been reported. In India, pigeonpea-infecting emaraviruses PPSMV-1 and PPSMV-2 are a major threat in most pigeonpea-producing regions where sterility mosaic disease is endemic. The disease was also reported in other parts of South-East Asia including Bangladesh, Nepal, Thailand, Myanmar, and Sri Lanka [108]. In China, AcCRaV and Actinidia virus 2 (AcV-2), both infecting kiwifruits, have been found in several provinces [35,39]. Further emaraviruses including JYMaV [40], PerMV [109], LiCRaV [47], Camellia chlorotic ringspot virus 1 (CaCRSV-1), and Camellia chlorotic ringspot virus 2 (CaCRSV-2) [44], PCLSaV [48], chrysanthemum mosaic-associated virus (ChMaV) [50,110], Japanese star anise ringspot-associated virus (JSARaV) [57], Vitis emaravirus (VEV) [53,54], alfalfa ringspot-associated virus (ARaV) [52], Ailanthus crinkle leaf-associated emaravirus (ACrLaV) [60], and PloAEV that infect important commercial fruits, ornamental crops, and trees have been described in China and Japan.

From other parts of the world, there is limited information on emaraviruses diversity and distribution. HPWMoV was reported from South America [111]. In Africa and the Middle East, FMV is present in many countries including Egypt and Tunisia [112,113], Turkey [114], Iran [115] and Saudi Arabia [116]. In Turkey, PiVB was identified in pistachio [37]. From Oceania, FMV [117], HPWMoV [98], karaka Okahu purepure virus (KOPV) [56], and ARaV [51] have all been reported.

In general, information on distribution is only well-known for emaraviruses that cause well-documented diseases that are major threats for commercial cultivation and export goods. This applies to diseases in fig, roses, pigeonpea, and cereals [10,65,66,118]. Globally, of close to 2000 investigated fig plants, a third tested positive for FMV [77]. International trade very likely contributes to the spread of emaraviruses over long distances, e.g., through trade of infected plant material. Considering that studies on emaraviruses are not carried out with the same intensity in all continents and surveys for newly detected emaraviruses have yet to be performed, information about the geographic distribution is still incomplete.

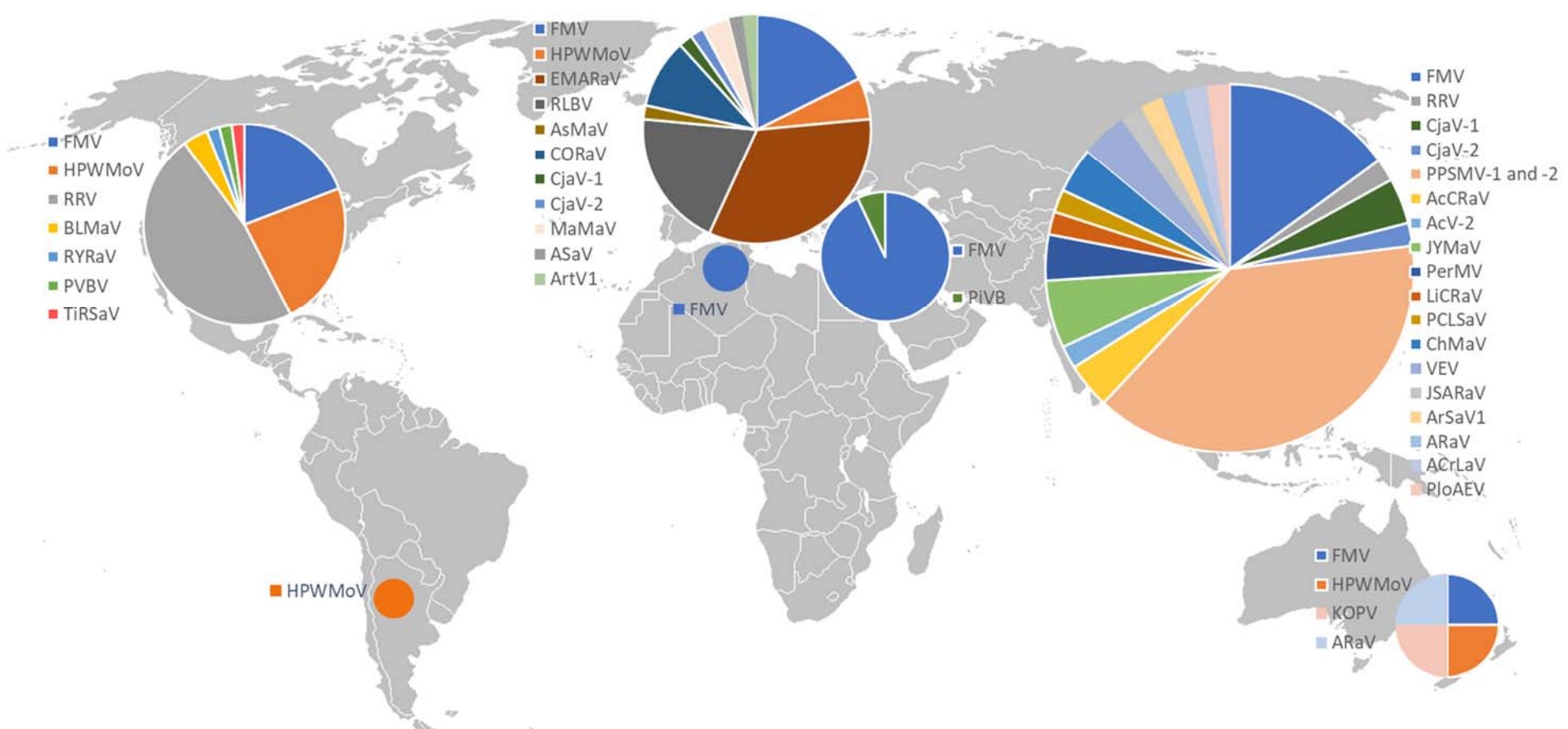

**Figure 4.** Worldwide distribution of emaraviruses as of 2022 estimated from referenced scientific publications. Emaraviruses detected in different parts of the world are illustrated as pie charts. Each sector represents a species and its proportion of the pie reflects the number of referenced articles. Detailed information is given in Table 1 and the main text.

## 5. The Four Main Phylogenetic Clades of Emaraviruses

Emaravirus species group into four main clades (A, B, C, and D) (Figure 5) when considering the amino acid (aa) sequences of the RNA-dependent RNA polymerase (RdRP, top left), the glycoprotein precursor (GPP, top right), the nucleocapsid (NC, below left), and the viral movement protein (MP, below right) [20,32,35].

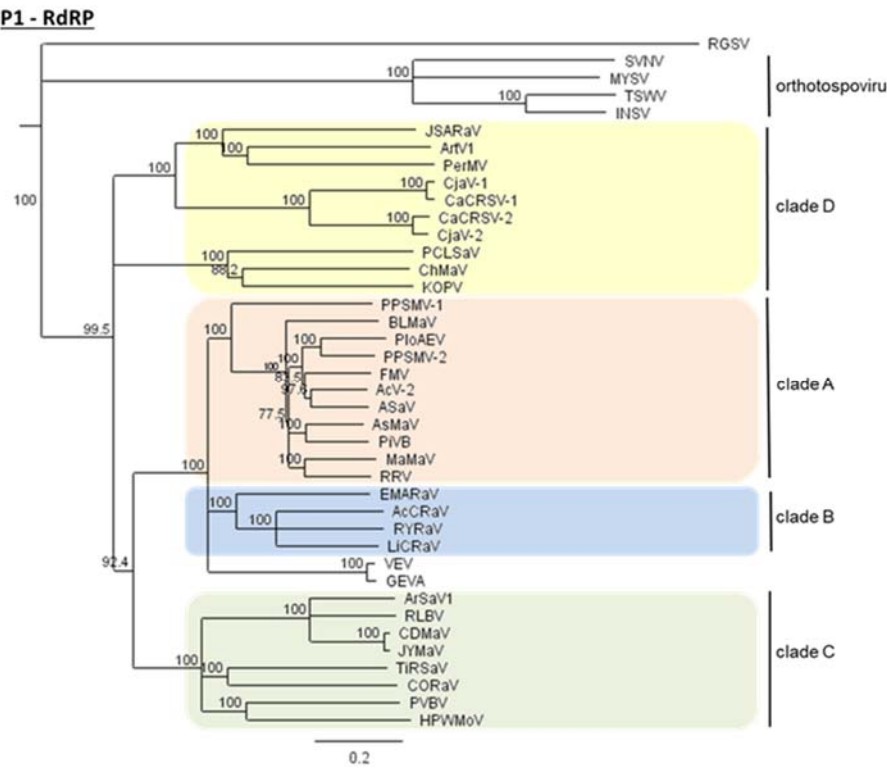

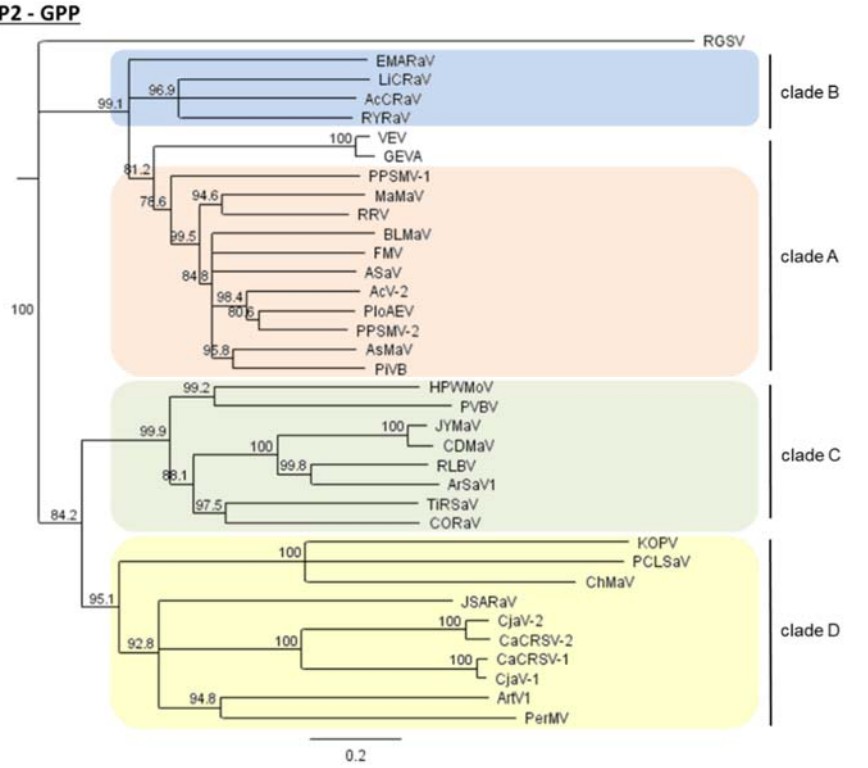

**Figure 5.** *Cont.*

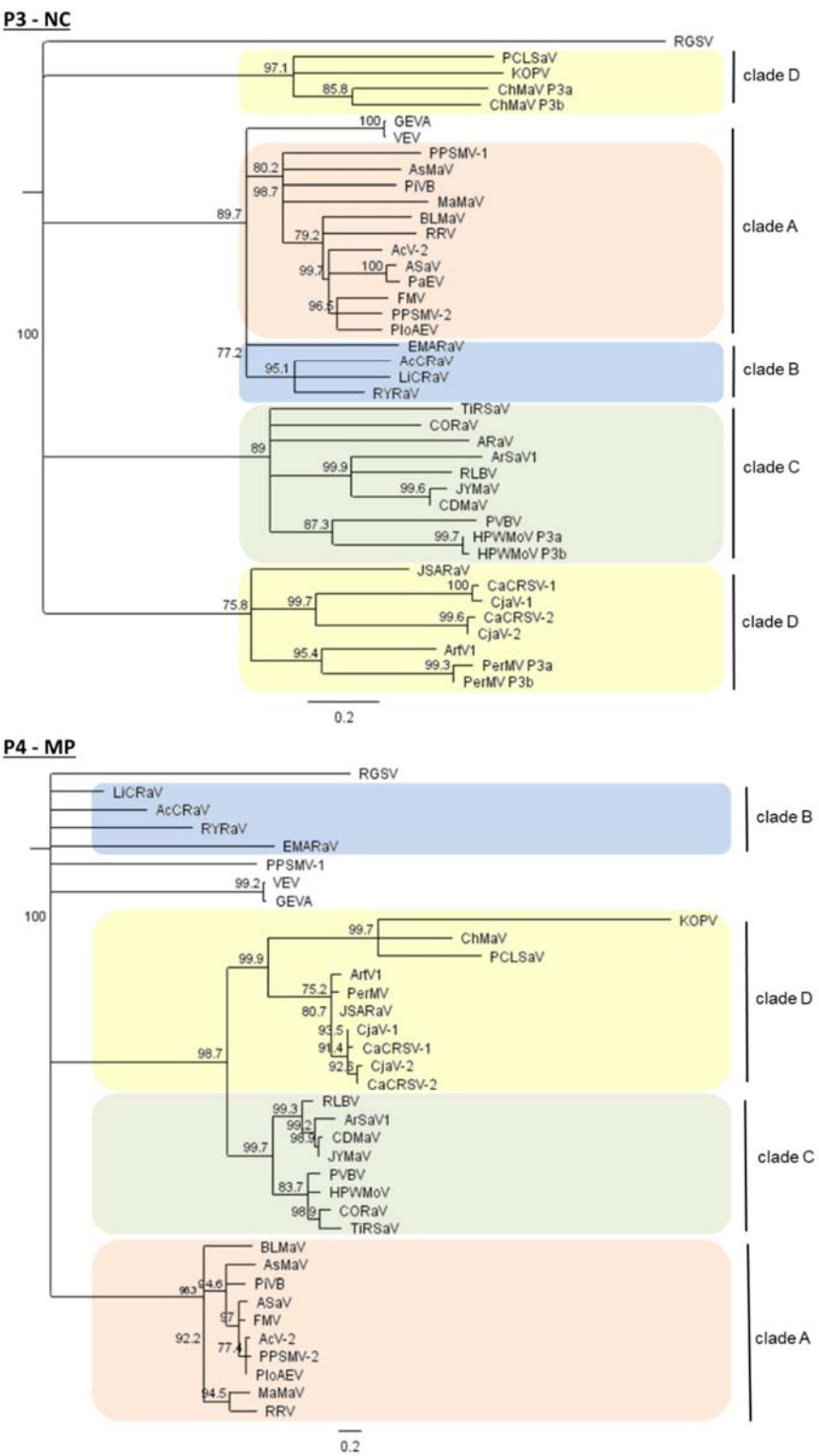

**Figure 5.** Emaraviruses cluster in four main clades. Amino acid (aa) sequences of emaraviral RdRP, GPP, NC, and MP were aligned using MUSCLE. Phylogenetic trees were built using neighbor-joining methodology in Geneious prime 2019.1.1. Abbreviations of virus names are explained in Table 1. Accession numbers used for the analysis are given in Supplemental Table S1. Branches report bootstrap support (1000 replicates). In the RdRP panel, rice grassy stunt virus (RGSV, *Tenuivirus*) was used as an outgroup, while the RdRP of selected orthotospoviruses was used to show the placement of emaraviruses within *Bunyavirales* (INSV: impatiens necrotic spot virus; virus; MYSV: melon yellow spot virus; SYNV: soybean vein necrosis virus; TSWV: tomato spotted wilt virus).

Clade A and B most probably have a common origin. They cluster together, with high bootstrap values supporting this phylogeny, when considering the RdRP, GPP, and NC which are relevant for the species demarcation criteria established by the ICTV. Notably, VEV cannot be grouped into one group, since it clusters either as a sister group of clade A (when we consider the GPP), or as a sister group of clades A and B (when we consider the RdRP and NC) (Figure 5).

Species belonging to clade C and D are more distantly related and, depending on the protein considered, either have or do not have a common ancestor. In clade D, two subgroups are detectable irrespective of the protein considered: (1) PerMV, JSARaV, Artemisia fimovirus 1 (ArtV1) and camellia-infecting species (CjaV-1 and CjaV-2), and (2) PCLSaV, ChMaV and KOPV (Figure 5). For the NC, these two subgroups even represent distinct groups within the genus *Emaravirus* (Figure 5, NC).

The taxonomical implications of emaraviral phylogeny are discussed in Section 7.

## 6. Genome Organization and Genetic Diversity

### 6.1. The Core Genome of Emaraviruses

The genome of emaraviruses consists of multiple single-stranded RNA segments, each containing a major open reading frame. A core of four genome segments is present in all members described so far. The core genome consists of RNA 1, coding for the RdRP, a GPP encoded by RNA 2, the NC encoded on RNA 3, and the RNA 4 coding for the viral MP (Figure 6). The four core segments of RRV were shown to be alone sufficient for the establishment of a systemic infection and the induction of symptoms in *N. benthamiana* [119].

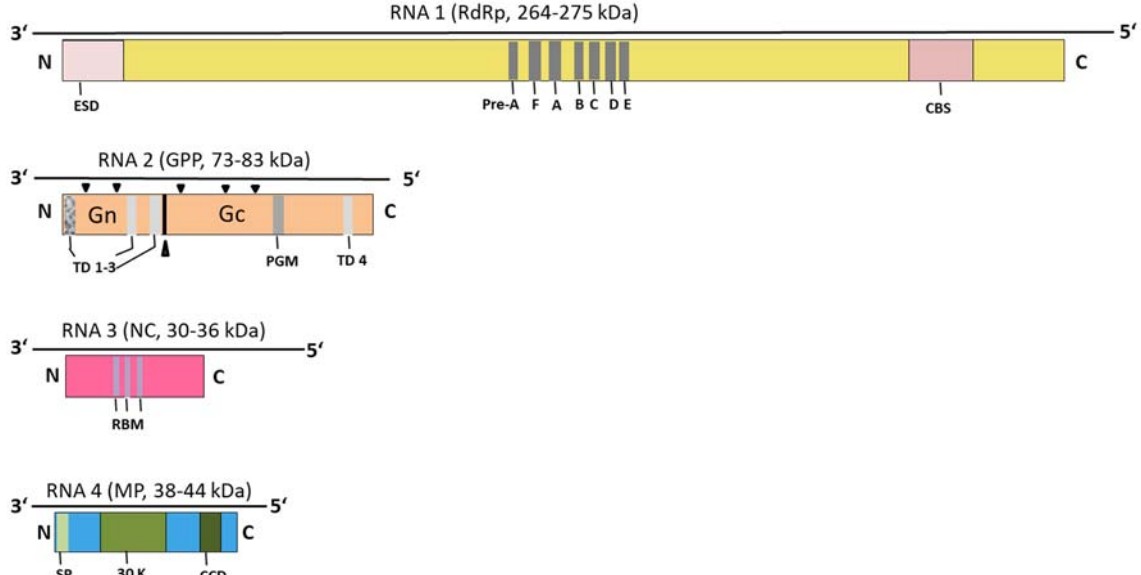

**Figure 6.** Core genome of emaraviruses. RNAs 1–4 encode the RNA-dependent RNA-polymerase (RdRP), glycoprotein precursor (GPP), nucleocapsid (NC), and movement protein (MP). Protein-coding regions (ORF) for each RNA are represented by colored boxes, as illustrated by the ICTV [1]. Protein motifs identified in P1–P4 are indicated: Pre-A, A, B, C, D, E, F—motifs of the central region of the RdRP; ESD—endonuclease sub-domain; CBS—cap binding site; TD—transmembrane region; PGM—phlebovirus glycoprotein motif; RBM–RNA-binding motifs; SP—signal peptide; 30 K–30 K superfamily domain; CCD—coiled coil region. Black arrows for GPP indicate either glycosylation sites (small arrows above), which vary in number and position between emaraviruses, or putative cleavage site (large arrow below) which cuts the GPP into a smaller (Gn) and a larger (Gc) glycoprotein.

RNA 1 is the largest genome segment with a length of ~7 kb, and encodes an RdRP of ~270 kDa. The polymerase contains seven motifs, named Pre-A, F, A, B, C, D, and E [111], of which six are also found in most RdRPs of the family *Peribunyaviridae* [120]. The

N-terminus of the RdRP contains a predicted endonuclease domain [121]. This domain is thought to play a role in "cap-snatching" (as in other *Peribunyaviridae* or *Orthomyxoviridae*), a mechanism in which the 5′ end of a host cell RNA is removed (snatched) and used as the 5′ cap and primer to initiate the synthesis of the nascent viral mRNA [122]. For FMV and RRV, there is experimental evidence that cap snatching is indeed used [26,123]. Accordingly, the sequence signature of a cap-binding site was detected in the C-terminal part of emaraviral RdRPs [25,26,122,123].

RNA 2 is the second largest segment, with a length of 2–2.5 kb, and encodes a glycoprotein precursor (GPP) of 73–83 kDa. It contains a predicted N-terminal signal sequence, transmembrane regions, and potential glycosylation sites (Figure 6), as well as a motif found in the glycoprotein precursors of phleboviruses (family *Peribunyaviridae*) [46,122]. By analogy with other *Bunyavirales*, the GPP is thought to be processed by a protease into two glycoproteins, a smaller N-terminal one (Gn), and a larger C-terminal one (Gc) (Figure 6). For PPSMV-1, a predicted cleavage site [FS↓DD] would yield two proteins of 22 and 52 kDa [122]. This site is also present in other emaraviral GPPs, with the exception of RLBV. For PerMV, a third cleavage site was predicted, which would yield a third short N-terminal peptide of 2.4 kDa [46]. Further cleavage sites have been postulated for other emaraviruses, such as RLBV [28]. However, none of the processing sites of the emaraviral GPPs have been experimentally confirmed.

RNA 3 is 1.1–1.6 kb in length and encodes a nucleocapsid protein (NC) of 30–36 kDa. This NC has significant sequence similarity with those of viruses belonging to the family *Peribunyaviridae* (detected using HHpred [124], our observations), which suggests they are likely to adopt very similar 3D structures. Sequence motifs conserved in P3 of FMV, EMARaV, and MRSV (syn. HPWMoV) were previously identified in the NC [125], which are thought to function as RNA-binding motifs [34]. Understanding their role should now be possible thanks to 3D structure prediction, which has become highly reliable [126–128]. Of note, HPWMoV [33], PerMV [46], PiVB [37], and ChMaV [50] encode two variants of P3.

RNA 4, being 1.1–1.6 kb in length, encodes the movement protein (MP, P4) of 38–44 kDa. It comprises a predicted signal peptide and a central domain homologous to the 30 K superfamily of plant virus MPs [129,130]. The function of the MP was confirmed experimentally: in RLBV, it localizes at the plasmodesmata [28], while in FMV, it complemented the cell-to-cell spread of a movement-defective potato virus X and formed tubule-like structures at plasmodesmata [130,131]. Some emaraviruses encode more than one MP. For example, JYMaV RNA 4 and RNA 5 each encode a MP, homologous to emaraviral MPs [40].

### 6.2. Most Emaraviruses Encode Additional (Non-Core) Genome Segments

Beyond the four core segments, most emaraviruses possess additional genome segments. The only exceptions are PVBV [38], CjaV-2 [43], ARaV [51,52], and ACrLaV [60], however, overlooked segments might await discovery (see Section 6.2.6). The number of additional segments differs considerably among species, from 1 for RYRaV to 6 for PerMV [46]. Our findings with respect to additional genome segments are summarized in Sections 6.2.1–6.2.6.

### 6.2.1. Full-Length RT–PCR as a Useful Approach to Detect and Characterize Emaraviral Genome Segments

Additional segments were identified by applying HTS or full-length RT–PCR with primers targeting the conserved 5′ and 3′ terminal sequences, which are shared both between genome segments and emaraviruses (see Section 6.3). For instance, a full-length PCR strategy was used to determine the first four complete genome segments of EMARaV [7] and later, a generic primer, PDAP213, was used to identify additional segments of RRV [27]. This primer was also used to amplify complete genome segments of other emaraviruses, such as RYRaV [34], EMARaV [14], AsMaV [17], CORaV [19], ASaV [21], and further species belonging to different clades. However, the RNA 1 with a size of more than 7 kb was usually not accessible using this approach. These successes illustrate how full-length

RT–PCR is a useful approach for detecting emaraviral genome segments and determining their sequence (Figure 7).

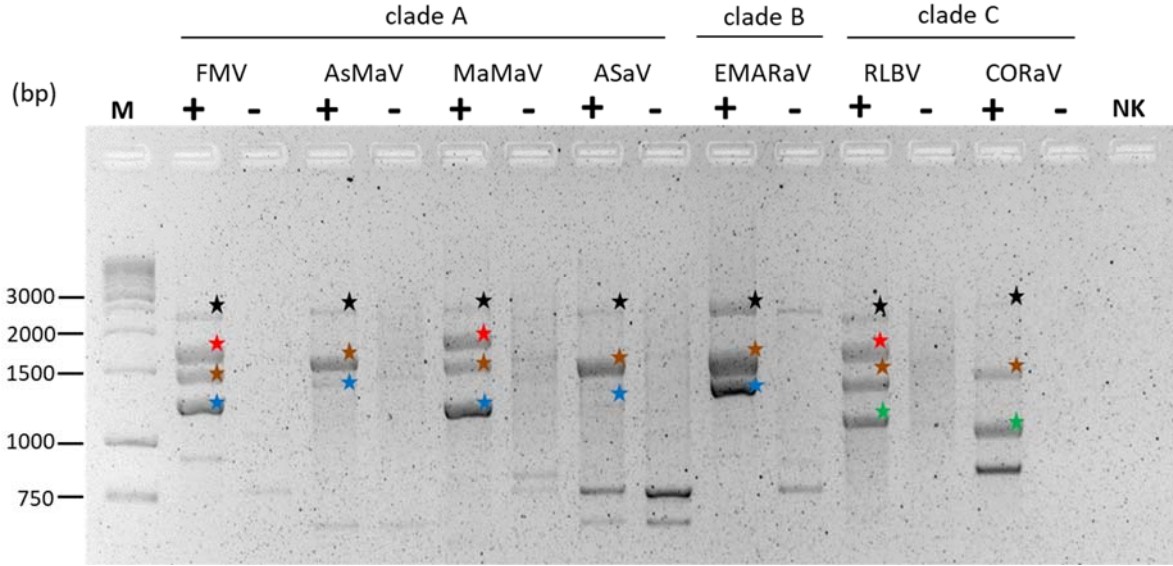

**Figure 7.** Full-length RT–PCR with generic primer PDAP213 targeting terminal 5′ and 3′ sequences for amplification of complete genome segments. PDAP213-primed cDNA from emaravirus-infected (+) and uninfected (−) leaf material was used for RT–PCR applying the terminal PDAP213 primer [27]. Using gel electrophoresis, full-length genome segments of emaraviruses can be detected including the core components (see Section 6.1) with the exception of RNA 1. The number and size of amplified segments depends on the species and the clade it belongs to. Clades are given above virus names. Fragments coding for homologous proteins are indicated with stars (★) in the same colors. Black indicates the GPP and brown indicates both the NC and MP, usually with similar length. Accessory genome segments are indicated by red, blue, and green stars (see below for details). Water was used as PCR negative control (NK). M–Gene Ruler 1 kb DNA Ladder (Thermo Scientific, Waltham, MA, USA).

6.2.2. Proteins Encoded by Non-Core Genome Segments Can Be Clustered into Three Main Groups

To classify proteins encoded by additional "non-core" segments, and in an attempt to predict their function, we performed advanced homology searches. These searches were both sequence-based, as described in von Bargen et al. [17] and Rehanek et al. [19], and structure-based, taking advantage of the revolutionary structure prediction software, Alphafold [126–128]. We identified, and report here, three groups of homologous proteins encoded by additional segments (Figure 8):

(1)   A protein of 460–500 aa, named P55 (in orange in Figure 8), found in some members of clades A, C, and D;

(2)   A protein of 170–250 aa, named "ABC" (in green in Figure 8), found in almost all members of clades A, B, and C;

(3)   A homolog of the *sadwavirus* Glu2–Pro glutamic protease (in yellow in Figure 8), found in some members of clades C and D.

Since this is a review, we only present here the minimum information required to classify emaraviral proteins into homologous groups and to guide further studies, rather than performing an extensive analysis.

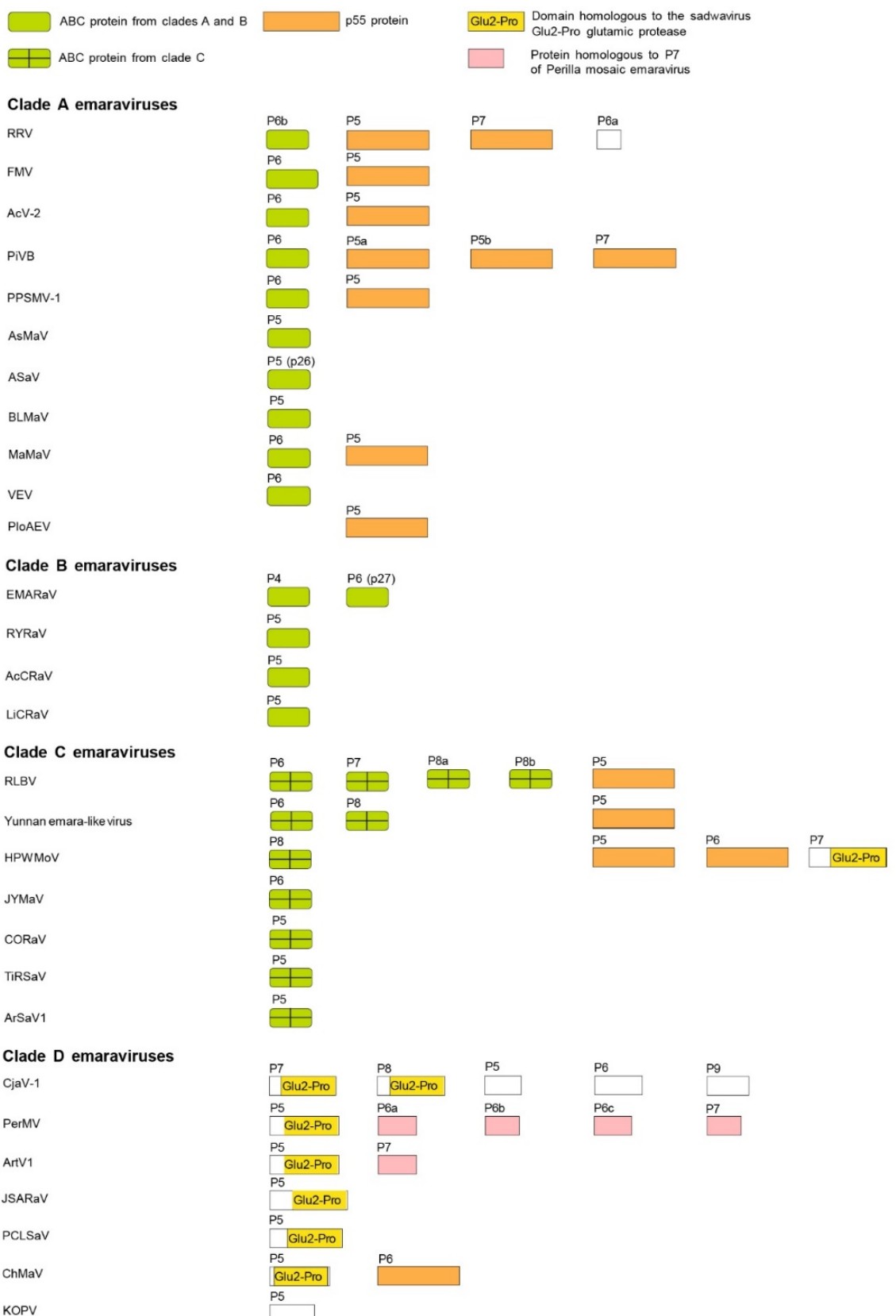

**Figure 8.** Emaravirus proteins encoded by non-core genome segments form three main groups of homologs. The names of proteins are those initially given in the first publication describing them and thus do not reflect homology relationships (see text). Abbreviations of virus names are explained in Table 1. Proteins are not represented to scale.

### 6.2.3. Some Emaraviruses in Clades A, C, D Encode a Protein of ~55 kDa, P55, Containing a Highly Conserved C-Terminal Domain

We report here that about half of emaraviruses encode a homolog of RRV P5, of comparable length (~450–500 aa, i.e., 49–55 kDa), which we named "P55". P55, represented in orange in Figure 8, is encoded mostly by members of clade A, three members of clade C, and only one member of clade D (ChMaV P6). It is absent in clade B members. Some emaraviruses encode several P55 homologs, with as many as three in PiVB (see Figure 8).

We predicted the structure of RRV P55 using Colabfold [132], a web-based implementation of Alphafold [127], which returned confident predictions (pLDDT > 0.70 for most regions). P55 is organized into two domains: a N-terminal domain (~aa 1–290 in RRV P5), henceforth called NTD, and a C-terminal domain, named CTD (~aa 327–460 in RRV P5). The NTD and CTD are separated by a long (30–50 aa), variable linker. Sequence analysis (not detailed here) indicates that all P55 proteins are composed of these same two domains. However, only in the CTD, conservation between P55 proteins is noticeable upon visual inspection. Figure 9 presents a sequence alignment of the CTD of all emaraviral P55 proteins, highlighting sequence conservation across species.

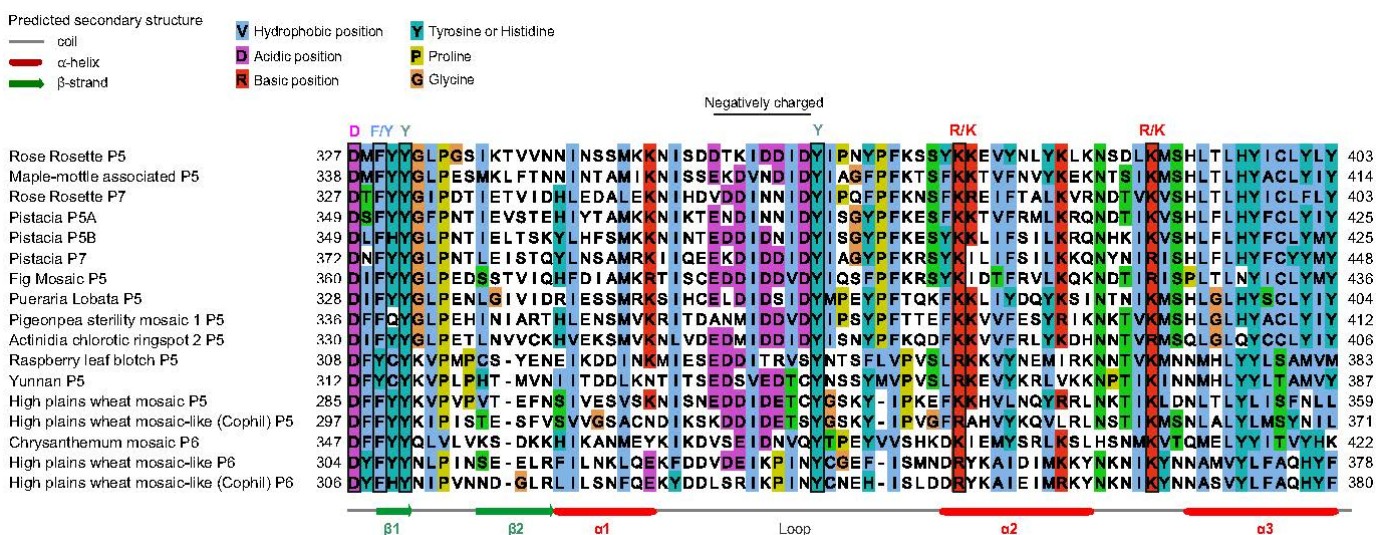

**Figure 9.** The CTD domain of emaravirus P55 proteins is highly conserved in sequence. Strictly conserved positions and semi-conserved positions are indicated above the alignment. Accession numbers are given in Table S2.

There are only two experimental clues regarding the localization and function of P55. First, RLBV P5 formed small aggregates in the cytoplasm, localized near the periphery of the cell, when expressed as a GFP fusion in *N. tabacum* [28]. Second, in RRV, P5, and P7 were dispensable for the successful transcription of a minireplicon [119].

### 6.2.4. Almost All Emaraviruses in Clades A, B, and C Encode a Homologous "ABC" Protein of 18–27 kD, Involved in Pathogenicity

We reported previously that:

- All emaraviruses in clade A and B encode at least one homolog of AsMaV P5, of 190–250 aa (21–27 kDa) [17];
- All emaraviruses in clade C encode a homolog of CORaV P5, of 165–230 aa (18–25 kDa) [19].

We report here that these two groups of proteins are in fact homologous and therefore all emaraviruses in clades A, B, and C encode a homologous protein of 18–27 kDa, which we named "ABC protein". It is colored green in Figure 8.

Colabfold predicted the fold of a representative of both types of ABC proteins (AsMaV P5, from clade A, and JYMaV P6, from clade C), with high confidence (pLDDT ≥ 0.90 for most regions). Thus, the prediction is expected to be very close to the actual 3D structure.

As seen in Figure 10, both proteins have significant structural similarity (*E*-value $E = 10^{-5}$ reported by the software mTM-Align [133], with an RMSD of 2.17 Å), indicating that they are homologous. Figure 10 presents a structural alignment of the two proteins, both including a core of 5 α-helices.

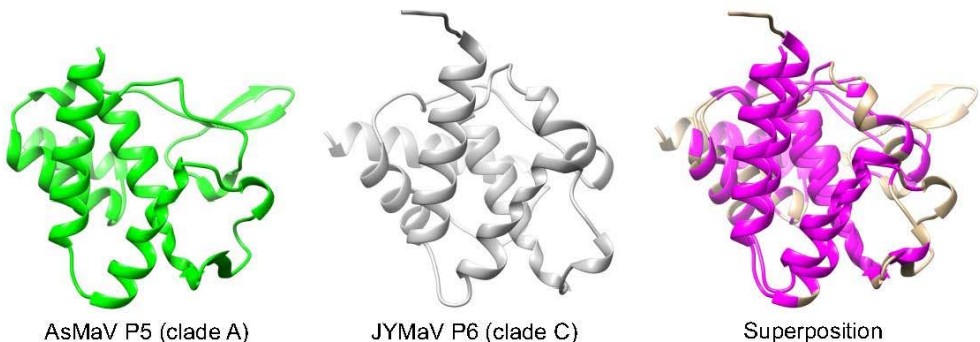

**Figure 10.** The 3D structures of ABC proteins from clade A and C are significantly similar, proving their homology. Superposition of the predicted 3D structures of aa 18–163 of AsMaV P5 and aa 7–139 of JYMaV P6. In the superposition panel, superposed regions (pairwise residue distance < 4 Å) are in magenta, while AsMaV P5 and JYMaV P6 are in green and grey, respectively.

Experimental data provides a few clues regarding the function of the ABC proteins. The ABC protein of HPWMoV, P8, was reported to have silencing suppressor activity [134,135]. In contrast, although the ABC proteins of RLBV, P6, P7, P8a, and P8b were reported to have no silencing activity, they do play a role in pathogenicity [29]. Thus, the ABC proteins appear to be involved in pathogenicity, and may act as silencing suppressors in some species. Interestingly, RLBV P6, P7, and P8 self-interact, which may be a general property of ABC proteins. Finally, in RLBV, P8 interacted with the nucleocapsid protein [29].

6.2.5. Some Emaraviruses from Clades C and D Encode a Homolog of the Sadwavirus Glu2–Pro Glutamic Protease

We report here that some emaravirus proteins from clades C and D contain a domain homologous to Glu2–Pro, a glutamic protease recently discovered in strawberry mottle virus (SMoV, genus *Sadwavirus*, family *Secoviridae*) [136]. We identified this domain by homology searches (using Psi-blast [137,138] and HHblits [139]) initiated from the *Sadwavirus* Glu2–Pro. The Glu2–Pro domain is colored in yellow in Figure 8. It is found in all members of clade D except KOPV, and in one member of clade C, HPWMoV (in P7). Finally, a Glu2–Pro domain in the P1 protein (also called "18.9 K protein") of a single tenuivirus, RGSV, was also identified.

Figure 11 presents an alignment of the Glu2–Pro domain of representative viruses. In most emaraviruses, the Glu2–Pro domain is preceded by an N-terminal extension of ~100–150 aa (not shown). Most positions strictly conserved in the Glu2–Pro of *Secoviridae* and *Closteroviridae* are also conserved in the Glu2–Pro of emaraviruses and of RGSV. In particular, the two glutamates (E) required for proteolytic activity in SMoV [136] are also strictly conserved. Therefore, the Glu2–Pro domain of emaraviruses and of RGSV can be expected to have catalytic activity, although its precise nature might differ from that of SMoV. Further research is required to discover its target(s).

Interestingly, a recent study [140] predicted that Glu2–Pro was structurally and functionally similar to another protease, Neprosin, found in plants, which cleaves proteins after prolines [141] like Glu2–Pro [136]. Both enzymes adopt the same predicted glucanase fold, and the two glutamates involved in catalysis in Glu2–Pro [136] are superposed to two conserved glutamates in the predicted 3D structure of Neprosin [140], suggesting that both enzymes have a similar catalytic mechanism. Emaravirus Glu2–Pro enzymes, which are shorter than other Glu2–Pro homologs (as short as 174 aa for ChMaV P5), may be of interest, since prolyl endoproteases have biotechnological and biomedical applications [142].

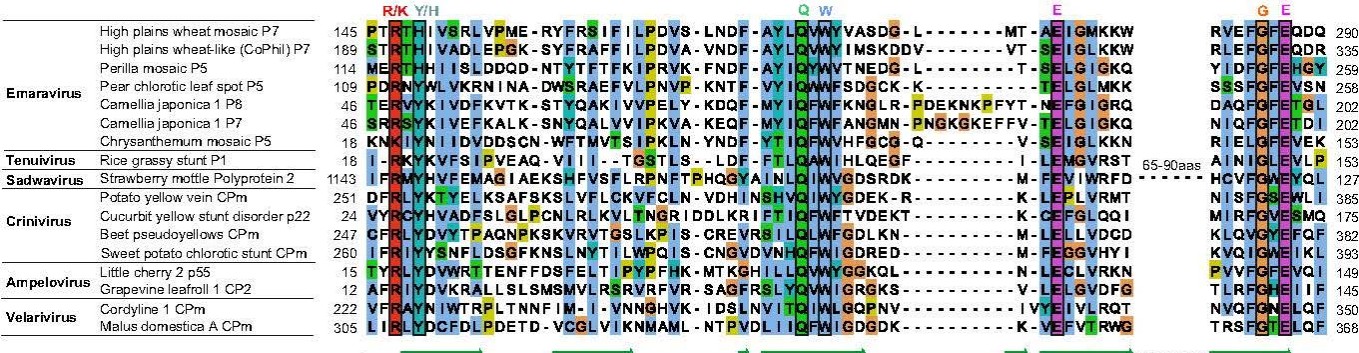

**Figure 11.** Glu2–Pro domain. Alignment of the Glu2–Pro domain from representative plant viruses. Conventions are the same as in Figure 8. Asterisks indicate the two strictly conserved glutamates (E) necessary for proteolysis in strawberry mottle virus. Accession numbers are given in Table S3.

Functional information is currently only available for one emaraviral Glu2–Pro: HPW-MoV P7, which acts as a silencing suppressor and delays the onset of the dsRNA-induced transitive pathway of RNA silencing [135]. P7 binds to dsRNA without size specificity and protects long dsRNAs from Dicer activity in vitro. Interestingly, HPWMoV P7 contains a GW dipeptide (aa 218–219), located in the Glu2–Pro domain (aa 147–288), whose mutation abrogated RNA silencing and enhanced pathogenicity [135]. (This GW is not visible in Figure 11 because it is located in a long region not conserved outside of emaraviruses and is thus not presented). The authors hypothesized that the GW dipeptide forms a "platform" that recruits the AGO protein by a mechanism similar to that of other silencing suppressors [143,144]. However, recent studies have shown that GW or WG motifs within AGO-binding platforms are usually only found in a very specific sequence context (frequent presence of other GW motifs, within structurally disordered regions that are enriched in small or charged aa [145], as reviewed in Ref. [146]). The single GW dipeptide of HPWMoV P7 is not located in such a context. AGOS, a predictor of AGO-binding platforms [146] calculates that it has a near random probability of occurrence (*p*-value = 0.5). Consequently, it seems unlikely that the GW dipeptide contributes to a traditional AGO-binding platform.

### 6.2.6. Species of Clade D Contain Non-Core Segments That Encode a Wider Variety of Proteins than in Other Clades

In clade D, non-core segments encode either Glu2–Pro, P55, or various "orphan" proteins without apparent sequence similarity to other proteins. We report here that five of these proteins are homologous, having weak but significant sequence similarity (detected using HHpred [124]). They are colored in pink in Figure 8: PerMV P7, P6a, P6b, and P6c, as well as the P7 protein of ArtV. (ArtV1 P7 is encoded by RNA 7, Genbank accession number OP441764.

### 6.3. Incomplete Genome Information and Harmonization of Proteins Names

Our investigations reveal two implications for future characterization of emaraviral genomes:

First, it is obvious that some genome segments might have been overlooked during viral characterization in some species, as was the case, for instance, for EMARaV. Although the virus was first described in 2007 [7] containing four genomic RNAs, the MP coding genome segment was identified 12 years later due to the advancement of molecular techniques [14]. A second example is PVBV (clade C), for which only the four core segments have so far been described [38]. Since almost all members of clade C encode at least another protein (P55, ABC and/or Glu2–Pro), it is plausible that additional genome segment(s) encoding at least one of these proteins remain to be identified. Of note, after this manuscript was completed, we recognized the sequence of "P8" of PVBV just being released (accession

number UWT50532.1). Blastp revealed similarities to P6 of JYMaV (clade C) confirming our hypothesis that PVBV contains an additional genome segment encoding an ABC protein.

Furthermore, even a passing look at Figure 8 suggests that an ABC homolog might have been overlooked in PloAEV (clade A), since the 10 other members of the clade encode an ABC protein. Likewise, a Glu2–Pro homolog might yet to be discovered for KOPV (clade D), since all other members of this clade encode at least one Glu2–Pro. Finally, a P55 homolog may have been overlooked in several species in clade A, since of the 11 species in this clade, seven encode at least one P55 protein. We put this hypothesis forward merely to guide future experiments, and make no claim regarding its likeliness.

Given their diversity and as indicated above, the best strategy to identify overlooked genomic segments is likely to be the use of primers targeting the conserved genome ends or, alternatively, the use of primers specific for each of the three groups of homologs we described here (P55, ABC, Glu2–Pro).

Given the pervasiveness of these additional genomic segments, the name of emaraviral proteins will need to be harmonized. At present, the numbering of emaravirus RNAs is not consistent because they were named in chronological order. Likewise, the name of the corresponding proteins does not reflect the relationships described here. For example, EMARaV RNA 4 does not encode an MP, as for the RNA 4 of other emaraviruses (Section 6.1), but rather an ABC protein (Figure 8). The MP of EMARaV was discovered in later studies and is encoded by a genome segment which was, and still is, called "RNA 5" [14]. A standardized naming and grouping of identified RNAs and encoded proteins according to the homologies presented would be much more preferable for comparative analyses.

*6.4. Terminal Ends of Genome Segments Are Conserved across Emaraviruses, with Variations in a Subgroup of Clade D*

Emaraviruses contain stretches of 13 conserved nucleotides at the 5′ and 3′ ends of each genome segment, which are partially complementary to each other [7]. They are 5′-AGUAGUGUUCUCC-3′ for the 5′ terminus and 5′-AGUAGUGAACUC for the 3′ terminus (ref. [26]; Figure 12). Due to their partial complementarity, panhandle structures are able to form for each RNA segment, resulting in a circular arrangement within the virus particle.

For all species identified so far, there is a conserved AGUAGU at position 1–6, conserved cytosine at position 10, and conserved uracil at position 11 for the 5′-terminus sequence, as well as a conserved ACUACU at the last six positions of the 3′-terminus. In addition, in all species of clades A, B, and C and in the subgroup of clade D comprising PCLSaV, ChMaV, and KOPV, there is strict conservation of a guanine at position 7 and two cytosine at positions 12 and 13 for the 5′-terminus.

For the 3′-terminus, these species contain a conserved cytosine at position 7 (counting from the 3′ end) and a conserved GGA at positions 11–13. One exception is ChMaV: in its RNA 2, positions 12 and 13 at the 3′-terminus contain two adenines, while they contain two guanines in RNA 1–4 and RNA 6.

In contrast, PerMV, JSARaV, and camellia- infecting species (CjaV-1 and CjaV-2) contain unique 11-nt and 12-nt sequences at the 3′ and 5′ termini, with a uracil present at position 7 at the 5′-terminus and specific nucleotides at the 3′-terminus conserved only across these three species [44,46]; Figure 12). This difference is in line with the separate clustering within clade D of PerMV, JSARaV, and CjaV-1 and CjaV-2 (Figure 5).

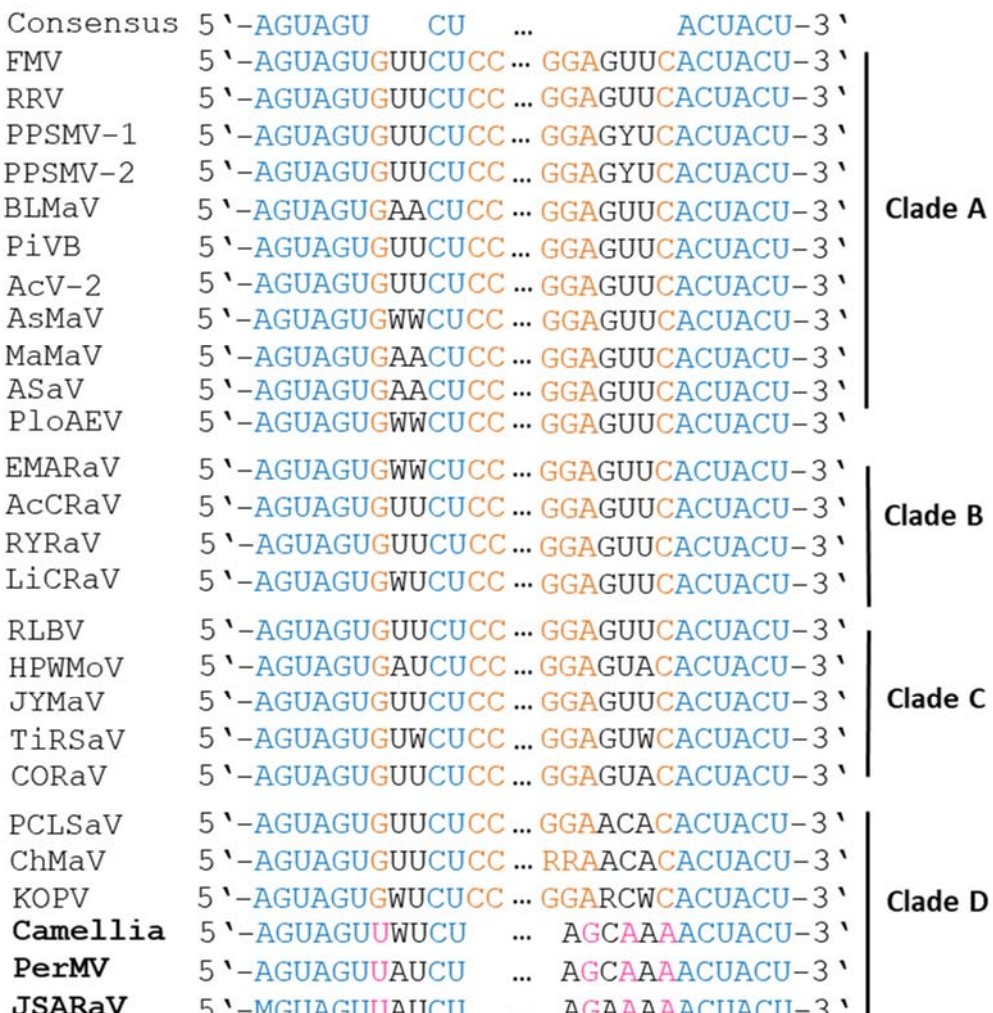

**Figure 12.** Emaravirus terminal genome ends. Nucleotide sequences at the 5′- and 3′-terminus of RNA segments of emaraviruses are shown, grouped by their phylogenetic placement (Section 5). Conserved nucleotides for species belonging to the genus *Emaravirus* are given above the figure and colored in blue. Further common nucleotides for emaraviruses except CjaV-1, CjaV-2, PerMV, and JSARaV (bold printed) are shown in orange. Unique nucleotides for camellia-infecting emaraviruses (CjaV-1 and CjaV-2), PerMV, and JSARaV are illustrated in pink. W = A/U; Y = C/U; R = A/G; M = A/C.

*6.5. Emaraviruses Show Significant Intraspecific Genetic Variability*

The extent of genetic variability observed for emaravirus isolates of different origins is dependent on the species and the genome segment considered. Genetic reassortment between genome segments, which promotes genome diversity was described for PPSMV-1 and PPSMV-2 as well as FMV [147,148].

Generally, the RNA 3 has been reported to be more variable compared to RNA 1, RNA 2, and RNA 4. Studies on isolates of PPSMV-1 and PPSMV-2 [148], and RLBV [149] revealed amino acid variability in the NC of between 8.5% and 9.3%, depending on the geographic distance of the populations [149,150]. Movement of infected plant material might explain this phenomenon, as suggested for FMV [147]. High expression levels of the P3 (the main structural protein required for packaging the emaraviral genomic RNAs) may favor the introduction of mutations. Thus, the pool of variants might facilitate the assembly of the viral ribonucleoprotein complex or confers selective advantages [46]. For example, differences in HPWMoV NC sequences were associated with different infectivity of isolates in maize [151].

On the other hand, evolutionary links between the virus, the host, and the vector might reduce the genetic variability of the NC for other emaraviruses as suggested by previous studies [69,152]. Studies on JYMaV revealed the RNA 3 is quite conserved with nucleotide identities of 98%–99% among isolates [153]. High sequence identities for the NC were also reported for different variants of RYRaV [34], RRV [26], PCLSaV [48], and ASaV [21]. Similarly, Roßbach et al. [107] reported a 97%–100% sequence amino acid identity when comparing the NC of 36 EMARaV isolates from Scotland, Sweden, Germany, Norway, and Finland which was supported by further studies [23,69,152,154]. Interestingly, six variants detected in Norway exhibited a much higher divergence level of 7.4% [107], which is comparable to the variation observed for the species mentioned above.

RNA 1 and RNA 2 are reported to be more conserved genome segments. For example, Steward et al. [155] identified more than 99% amino acid sequence identity for the RdRP for three isolates of HPWMoV. Liu et al. [48] reported nucleotide sequence identities of between 90%–95% for the protein-coding region of RNA 2 among three PCLSaV isolates. High MP sequence identities were also reported for RRV, PPSMV-1 and PPSMV-2, FMV, and PCLSaV [26,48,121,147].

There is little information concerning the diversity of non-core genomic components. High sequence identities were reported for the P4 of EMARaV which corresponds to ABC homology group [107]. RNA 6 of JYMaV was reported to be the most divergent among all genome segments with high sequence diversity in their 3' UTR [153]. In RLBV, the two RNA 8 segments encode highly variable proteins [29]. However, comparisons of ASaV variants from geographically distant locations identified high sequence identities for the partial RNA 5 at the nucleotide and aa level [21].

It is well known that RNA viruses have a high potential for genetic diversity, shaped by factors such as selection pressure, mode of transmission, host resistance, and others [156]. We are still far from knowing the mechanisms that drive the sequence variability of emaraviruses.

## 7. A Proposal: Dividing the Current Genus Emaravirus into At Least Two Genera

We propose to divide the genus *Emaravirus* into at least two genera, based on phylogeny, proteins encoded, and genome terminal ends (presented respectively in Sections 5, 6.2 and 6.3). One genus would comprise of clades A and B, and the other of clades C and D. When more sequences become available, clades C and D may need to be separated into two (or more) subgenera. We discuss our reasoning below.

The rationale for grouping clades A and B into a genus is strong. Clades A and B are monophyletic when considering the RdRP, GPP, and NC (Figure 5). In addition, their non-core proteins are very different to those encoded by emaraviruses representing clade C and D in two major ways. Firstly, the ABC proteins of clades A and B have significant sequence similarity to each other [17], but not to those of clade C (clade D does not encode an ABC protein). Secondly, in clade A or B, no segment encoding a Glu2–Pro protein has been reported (Figure 11), unlike in clades C and D. Therefore, clade A and B clearly form a separate phylogenetic unit, which would, in our opinion, justify the creation of a separate genus.

It is less clear whether clades C and D should each form an additional genus, or two subgenera grouped within a unique genus. Clades C and D are clearly distinct from clades A and B and appear monophyletic when considering the GPP and MP (Figure 5). Clade D is further differentiated from clade C by the apparent lack of a P55 protein (Figure 11), unlike some members of clade C. In fact, clade D itself might form two subgenera, since its species consistently cluster into two groups (Figure 5), with one group comprising PerMV, JSARaV, CjaV-1, and CjaV-2 having genome termini different from those of all other emaraviruses (Section 6.3).

In conclusion, only an in-depth analysis of emaravirus genome segments, and perhaps the discovery and sequencing of new members of clades C and D can help settle their taxonomy.

## 8. Established and Putative Members of the Genus

Additional emaraviruses have recently been detected and characterized. They share similarities with currently accepted species and are introduced in the following Section.

### 8.1. Established Emaraviruses

Currently accepted emaraviruses: (i) are multipartite, monocistronic, negative-sense, single-stranded RNA viruses with at least four genome segments encoding a RdRP, GPP, NC, and MP, respectively; (ii) contain conserved terminal ends capable of forming pan-handle structures because of their partial complementary; (iii) are visible as spherical enveloped particles with a diameter of 80–100 nm; (iv) are transmitted by eriophyid mites; (v) have the capability to use cap-snatching for transcription initiation [1]. According to these characteristics, the genus currently comprises 24 species [157]. Six of them (EMARaV, FMV, RRV, RLBV, HPWMoV, PPSMV) were reviewed by Mielke-Ehret and Mühlbach [8]. Eighteen others were recognized later and are therefore not covered by this early review. We have briefly summarized their main characteristics below.

### 8.1.1. Pigeonpea Sterility Mosaic Virus 2

A second emaravirus that is associated with SMD was identified across India's pigeonpea fields [32,148]. PPSMV-2 contains six genome segments and is phylogenetically distantly related to PPSMV-1. However, mixed infections as well as genetic reassortment between the two viruses, in the case of RNA 4, have been demonstrated [148].

### 8.1.2. Blackberry Leaf Mottle-Associated Virus

BLMaV was identified in blackberry plants suffering from the blackberry yellow vein disease which comprises of symptoms including leaf mottling, chlorotic ringspots and curved midribs. It is another and the second emaravirus infecting *Rubus* spp. and it contains five genome segments. BLMaV is transmitted by an eriophyid mite that remains to be determined [36]. BLMaV and RLBV are distantly related with BLMaV belonging to clade A and RLBV to clade C.

### 8.1.3. Actinidia-Infecting Emaraviruses

In kiwifruit, two distinct emaraviruses have been reported from China.

Actinidia chlorotic ringspot-associated virus (AcCRaV), the first emaravirus reported in kiwifruit is associated with ringspots, vein yellowing, and chlorotic spots in infected plants. The virus contains five genome segments and was detected in different kiwifruit species. Double-membrane-bound bodies resembling emaravirus particles were observed in infected tissues [35].

Actinidia virus 2 (AcV-2), the second emaravirus in kiwifruit reported, consists of six genome segments and is associated with leaf mottle, mosaic, and chlorotic spot symptoms [39]. Three *Actinidia* species including *A. chinensis*, *A. delicious*, and *A. eriantha* were found to be infected with AcV-2, the same species that were shown to be susceptible to AcCRaV. AcCRaV was additionally detected in *A. kolomikta* and some unidentified *Actinidia* spp. Although both viruses occur in the same regions, co-infection has not been found.

### 8.1.4. Redbud Yellow Ringspot-Associated Virus

RYRaV is reported from redbud plants displaying vein clearing, chlorotic ringspots, and oak-leaf pattern. The virus is graft transmissible and contains five genome segments. RYRaV is closely related to EMARaV, AcCRaV, and LiCRaV, in phylogenetic clade B [34].

### 8.1.5. Pistacia Virus B

Identified in Turkish pistachio trees, Pistacia virus B contains seven genome segments. It belongs to phylogenetic clade A and is most closely related to AsMaV [37].

### 8.1.6. Palo Verde Broom Virus

In blue palo verde trees located in the southwestern US and Mexico, PVBV was identified and found to be associated with witches' broom disease. For PVBV, only the four core genome segments are known. PVBV belongs to phylogenetic clade C with highest similarities to HPWMoV [38].

### 8.1.7. Jujube-Infecting Emaraviruses

Jujube yellow mottle-associated virus (JYMaV) was detected in jujube trees in China displaying the jujube yellow mottle disease. JYMaV comprises of six RNAs and phylogenetic analysis revealed its clustering within clade C emaraviruses [40]. Viral particles were documented in symptomatic leaf tissues by transmission electron microscopy.

In addition, Liu et al. [41] reported the discovery of Chinese date mosaic-associated virus (CDMaV) from other diseased jujube trees. Five RNAs were detected and *Epitrimerus zizyphagus* has been suggested as a vector candidate.

Since amino acid sequences of the RdRP, GPP, NC, and MP show very high sequence identities (each above 90%), JYMaV and CDMaV represent isotypes of the same virus, which is also supported by the phylogenetic trees (Figure 5).

### 8.1.8. Ti Ringspot-Associated Virus

TiRSaV has been discovered in ringspot diseased ti plants from Hawaii. It contains five genome segments and was shown to be transmissible to other experimental host plants by mechanical means [42].

### 8.1.9. Perilla Mosaic Virus

PerMV was detected in shiso plants displaying mosaic symptoms. Transmission by the perilla rust mite (*Shevtchenkella* spp.) has been demonstrated. PerMV belongs to phylogenetic clade D and possesses ten genome segments, the highest number of genome components known to date for emaraviruses [46,110].

### 8.1.10. Aspen Mosaic-Associated Virus

In diseased aspen trees from Fennoscandinavia, the five segmented AsMaV has been described. It was shown to cause the mosaic-disease of Eurasian aspen by graft transmission. It induces mosaic and mottle in leaves of aspen rootstocks grafted with mosaic-diseased, AsMaV-infected aspen scions [17]. RT–PCR and HTS analyses (RNASeq, Illumina, Illumina Solutions, Berlin, Germany) of pooled symptomatic leaf samples taken from the infected scions and rootstocks only detected AsMaV, therefore confirming the virus ability to cause the mosaic-disease. So far, the virus and the disease are only known to occur in Sweden, Finland, and Norway, affecting *P. tremula* [17]. The virus could not be detected in other *Populus* species until now (*P. nigra* and hybrids of this species) originating from random sample surveys in Germany (unpublished results).

### 8.1.11. Lilac Chlorotic Ringspot-Associated Virus

LiCRaV was identified in diseased lilac plants. It contains five RNA segments and clusters with clade B emaraviruses. Mechanical transmission to *N. benthamiana* was demonstrated [47]. Its precise relationship to symptoms in lilac remains to be established.

### 8.1.12. Pear Chlorotic Leaf Spot-Associated Virus

In leaf spot-diseased material from sandy pear, double-membrane bound bodies could be detected. PCLSaV contains five RNAs and belongs to phylogenetic clade D [48]. As for LiCRaV, precise relationships to symptoms are not established.

### 8.1.13. Camellia-Infecting Emaraviruses

In studies exploring the virome of diseased *Camellia japonica* in Italy and China, two novel emaraviruses were identified [43,44]. Camellia japonica-associated virus 1

(CjaV-1) and Camellia japonica-associated virus 2 (CjaV-2) are reported from camellia leaf material collected from nurseries in the Piedmont region in Italy. Nine and four genome segments were identified for CjaV-1 and CjaV-2, respectively [43]. In Chinese camellia plants displaying similar ringspot and color-breaking symptoms, CjaV-1 was also described [45].

In parallel, Zhang et al. [44] identified viral contigs related to two emaraviruses in camellia leaf material from the Jiangxi province of China displaying chlorotic ringspots. The viruses were named Camellia chlorotic ringspot virus 1 (CaCRSV-1) and Camellia chlorotic ringspot virus 2 (CaCRSV-2), with each containing five genome segments. Comparisons of the RdRP, GPP, NC, and MP revealed very high amino acid identities (above 90% for each protein considered; data not shown) between CjaV-1 and CaCRSV-1 as well as between CjaV-2 and CaCRSV-2. Similarly, phylogenetic analyses indicate that CjaV-1 and CaCRSV-1 represent different isolates of the same virus, which also applies for CjaV-2 and CaCRSV-2 (Figure 5).

### 8.1.14. Common Oak Ringspot-Associated Virus

In ringspot-diseased common oak (*Quercus robur* L.), CORaV was first identified in a seed orchard in Germany [18]. It contains five genome segments and is graft-transmissible to young oak seedlings [4]. Following sampling of leaf material from diseased oaks in different European countries, the virus was identified at several locations in Germany, Sweden, and Norway using specific RT–PCR-based detection. Virus detection is strongly associated with chlorotic ringspot symptoms [19]. CORaV also affects the popular variety 'Fastigiata koster' but could not be detected in other *Quercus* species to date. It clusters with clade C emaraviruses.

### 8.1.15. Maple Mottle-Associated Virus

In maple trees exhibiting mottle, chlorotic spots and mosaic on leaves, the six RNAs of MaMaV were detected. The virus was so far only confirmed in diseased maple trees in one location in Berlin Grunewald, Germany. It is closely related to clade A emaraviruses [20].

### 8.1.16. Chrysanthemum Mosaic-Associated Virus

ChMaV was identified in chrysanthemum plants with mosaic and chlorotic ringspot leaf symptoms. It contains seven genome segments and is closely related to clade D emaraviruses. *Paraphytoptus kikus* is a suspected vector [50,110].

### *8.2. Putative Emaraviruses*

Recently, further tentative emaraviruses have been reported (see Table 1). Given the species demarcation criteria for emaraviruses defined by the ICTV [1], which are (1) Differences in relevant gene product sequences of more than 25% for RNA 1-RNA 3; (2) Differences in host range, and (3) differences in vector specificity, these viruses will likely lead to the creation of additional species in the genus.

### 8.2.1. Alfalfa Ringspot-Associated Virus

In a survey of Australian alfalfa populations, ARaV was identified in South Australia and Victoria in plants displaying ringspot-like symptoms. In phylogenetic analysis, ARaV clusters with clade C emaraviruses [51] (Figure 5).

### 8.2.2. Ash Shoestring-Associated Virus

In diseased ash trees from Switzerland and Germany, ASaV with its five genome segments was identified. Using a specific RT–PCR assay the virus was detected in common ash (*Fraxinus excelsior*) and manna ash (*Fraxinus ornus*) in additional locations in Germany and Switzerland, as well as in Northern Italy and Southern Sweden [21].

Investigations of different German pea producing regions identified a novel virus, which was initially named pea-associated emaravirus. The virus was detected in two samples from Saxony showing chlorosis symptoms [55]. Interestingly, phylogenetic analysis

and alignments of the partial RNA 3 of the emaravirus reported from peas viral genome to other members of the genus revealed nearly perfect sequence identities to ASaV [21] (Figure 5, P3). Thus, AsaV appears to infect both ash and pea. It belongs to phylogenetic clade A.

### 8.2.3. Karaka Okahu Purepure Virus

In New Zealand, KOPV was detected in karaka trees (*Corynocarpus laevigatus*) showing chlorotic spots. KOPV contains five genome segments and is closely related to clade D emaraviruses [56].

### 8.2.4. Emaraviruses in Grapevine

In Asia, two emaraviruses have been detected in grapevine plants.

Nabeshima and Abe [53] identified Vitis emaravirus (VEV) in symptomless wild *Vitis coignetiae* plants from Hokkaido island Japan. VEV contains five genome segments and is related to clade A emaraviruses. (Figure 5).

In a nursery in Liaoning (China), Fan et al. [54] detected grapevine emaravirus A (GEVA) on the grapevine cultivar "Shennong Jinhuanghou" (*Vitis vinifera* L.) showing chlorotic mottling symptoms. GEVA contains five RNAs and could be graft-transmitted to healthy grapevines. Like VEV, GEVA is closely related to clade A emaraviruses.

Comparisons of the RdRP, GPP, NC, and MP amino acid sequences revealed very high sequence identities (above 93%) between VEV and GEVA. VEV and GEVA have therefore to be considered as synonymous and belonging to the same virus species.

### 8.2.5. Japanese Star Anise Ringspot-Associated Virus

JSARaV was detected in ringspot-diseased Japanese star anise (*Illicium japonicum* L.). The virus contains five genome segments and is putatively transmitted by an eriophyid mite of the family *Diptilomiopidae*. In phylogenetic analysis, JSARaV clusters with clade D emaraviruses [57].

### 8.2.6. Arceuthobium Sichuanense-Associated Virus 1

In *Arceuthobium sichuanense*, a parasitizing mistletoe on several species of the genus *Picea* (*Pinaceae*), ArSaV1 containing five RNAs was identified in southern parts of China. The virus is closely related to clade C emaraviruses [58].

### 8.2.7. Artemisia Fimovirus 1

In a large HTS-based approach to explore the virus diversity in tomato and weed fields in Slovenia, Artemisia fimovirus 1 (ArtV1) with its five genome segments was identified in *Artemisia verlotiorum*. It shows closest relationship to PerMV in phylogenetic analysis [59].

### 8.2.8. Ailanthus Crinkle Leaf-Associated Emaravirus

ACrLaV was identified in crinkle-diseased trees of heaven in China. It contains four genome segments showing highest similarities to clade C emaraviruses [60].

### 8.2.9. Pueraria Lobata-Associated Emaravirus

In kudzu from China, five complete RNAs of PloAEV were identified. Phylogenetic analysis revealed closest relationship of the virus which is mechanically transmissible to *N. benthamiana* with clade A emaraviruses [61].

In metagenomic datasets, sequences have been reported that indicate additional emaraviruses that may be present in further plant species. As only partial sequence information is available and de facto no biological features or characteristics are connected with these orphan sequences, we summarize them briefly in this paragraph.

### 8.2.10. Woolly Burdock Yellow Vein Virus

Using deep-sequencing of small-RNAs, sequence contigs related to emaraviruses were identified in wild woolly burdock plants from southern Finland displaying virus-like symptoms of vein yellowing and leaf mosaic. Deduced amino acid sequences of the partial nucleocapsid showed similarities to emaraviruses [62]. However, no information on additional genome segments, phylogenetic analysis, or other experimental evidence is available. Thus, the classification of the virus, named woolly burdock yellow vein virus (WBYVV) in the genus *Emaravirus*, remains speculative.

### 8.2.11. Yunnan Emara-like Virus

In a metatranscriptomic approach investigating the virus diversity in different environmental samples, partial sequence information of seven RNAs of a putative emaravirus were identified from cattle faeces obtained in the province Yunnan (China) [63]. The authors reported closest relation to RLBV (RNA 1-RNA 6 and RNA 8) in Diamond blastX from 34.3% (partial P6) to 72.3% (partial NC) on amino acid level suggesting a novel species in the genus related to clade C. However, as no additional biological information for instance on host species or confirmation of sequences are reported, further studies are required to confirm this novel virus as a plant-infecting emaravirus.

The NCBI database further provides sequence information for unclassified putative emaraviruses including:

- partial sequence of RNA 1 of Illicium anisatum ringspot-associated virus
- sequence for two different RNA 1 similar to established emaraviruses, identified in wild potato species in South Africa [64]

The accession numbers of these orphan sequences are listed in Table S4.

## 9. Diagnosis and Control Strategies

### 9.1. Diagnosis Needs

Reverse transcription–polymerase chain reaction (RT–PCR) has proven a very valuable tool for RNA virus detection and is widely used for the routine detection of several emaraviruses (Table 2). However, studies revealed differences in the efficiency of established RT–PCR tests for the detection of genomic RNAs, likely explainable by the sequence variability observed for most emaraviruses, as described in Section 6.4. Additionally, uneven distribution of emaraviruses in plant organs, tissue, variation of virus concentration over time, and differences in concentration of different RNA segments [20] may also contribute to difficulties with RT–PCR assays. This can be overcome by targeting more than one genome segment of these multipartite viruses in PCR-based assays [18,28]. Also, Sanger-sequencing of the amplified PCR product is advisable for the reliable identification of the virus species. In addition to species-specific primer pairs, Elbeaino et al. [158] developed degenerated genus-specific primer pairs targeting three highly conserved amino acid motifs within the RdRP-encoding RNA 1. RT–PCR using these primer pairs enabled the detection of emaraviruses infecting a wide range of hosts. However, not all known emaraviruses are detectable using these generic primer pairs, which are not as sensitive as species-specific primers (ref. [158] and own unpublished results). Due to this, and the increased number of emaraviruses described in recent years, Kubota and coworkers [110] designed improved degenerate primer pairs that were shown to detect a wider range of emaraviruses, including emaraviruses of all four main clades.

Although RT–PCR is widely used for emaravirus detection, further progress in the development of accurate diagnostic assays is required. We think this should be a priority for five reasons: (1) the ongoing dissemination of emaraviruses worldwide; (2) their detrimental effects on plant health; (3) the existence of latent infections; (4) our incomplete knowledge of emaravirus transmission modes; and (5) limited availability of serological detection methods. We discuss this below.

### 9.1.1. Worldwide Dissemination

The increasing international trade and exchange of numerous plant materials promotes the dissemination of viruses. Three examples highlight this fact. First, FMV was introduced in fig-growing areas worldwide, mainly by infected plant material [77]. Second, RRV was recently found in India, the first report from a country outside North America [103]. The virus and its vector were added to the A1 alert list by the European and Mediterranean Plant Protection Organization (EPPO) and the UK and RRV was listed as a quarantine pest in Morocco [159,160]. Third, the EU is currently planning to categorize HPWMoV as a potential Union quarantine pest due to its wide distribution in America, Australia, and the Ukraine [97,161].

### 9.1.2. Detrimental Effect on Plant Health

Emaraviruses can have significant effects on plant health of diverse plant species within agricultural, forest, and urban environments. That includes decreases in fruit quality and yield, leading to reduced income for crop producers [162]. We provide five examples. First, crop yield reductions attributed to PPSMV-1 and PPSMV-2 were reported in India, where pigeonpea is cultivated as a major protein source for millions of people [11,30]. Second, in the USA, there are tremendous yield losses for farmers growing wheat and maize on HPWMoV-affected fields, with partial crop failure [99]. Third, BLMaV infection of *Rubus* dramatically decreases the fitness of blackberry plants, by inducing blackberry yellow vein disease (BYVD). Disease-free plants can be productive for more than 20 years, whereas diseased plants become unproductive and uneconomical, leading to replanting 5–7 years after the onset of the disease [70]. Fourth, rose rosette disease, the most feared disease for roses, causes major damage and loss of profit for farmers in the USA [163,164]. Fifth, studies on trees of forests and urban stands reported reduced growth and decline over the years as seen in *Sorbus aucuparia* [6,165].

### 9.1.3. Latent Infections

In certain cases, symptoms caused by emaraviruses can be detected by visual inspection of leaves. However, the symptomatology of emaraviruses is complex, with latent (symptomless) infection reported for several emaraviruses [21,53]. In addition, the time between emaravirus infection and the appearance of symptoms is not yet clear. Plants with a latent infection might spread the virus unnoticed. Therefore, relying solely on symptoms should be avoided.

### 9.1.4. Incomplete Knowledge of Transmission Modes

As long as we have an incomplete picture of the modes of emaravirus transmission, virus spread cannot be completely prevented. As an example, seed transmission has yet to be proven convincingly. Tree nurseries rely heavily on conservation seed orchards, which provide seedlings and seeds to maintain resources of important deciduous tree species. In a screening of more than 1500 seedlings obtained from CORaV-infected mother trees, a single seedling displaying chlorotic ringspot symptoms tested positive for the virus [91]. To ensure the provision of certified seeds, testing of donor trees and seeds with reliable test systems is crucial.

### 9.1.5. Limited Availability of Serological Detection Methods

Serological detection of emaraviruses is rarely performed owing to a lack of virus-specific antisera for many species (Table 2). Antibody-based assays are available for economically important emaraviruses including FMV, HPWMoV, RRV, PPSMV, and EMARaV. Further test systems with antibodies against urban and forest tree emaraviruses will be available step by step in the near future.

Both PCR-based and serological tests are very sensitive and able to detect emaraviruses even at low concentrations, e.g., in their (putative) vectors or in dormant plant tissues outside the growing season. However, not every plant material is suitable for these detection

methods. The factors that contribute to difficulties in emaravirus detection are: (1) a low overall virus titer in some host plants, for example RRV [119]; (2) unequal distribution of the virus within its host [166]; (3) varying abundancy of genome segments in the plant tissue [17]; and (4) technical difficulties, e.g., high rates of secondary metabolites in some hosts and tissues, which interfere with RNA extraction and subsequent processing steps. To ensure virus detection, leaf material usually serves as the best starting material, but buds, fruits, flowers, or roots are also suitable.

**Table 2.** Published detection assays for emaraviruses.

| Species | Detection Assay | Reference(s) |
|---|---|---|
| Genus-specific | RT–PCR targeting RNA 1 pre-motif A, motif A, and motif C<br>RT–PCR targeting RNA 1 motifs F, A, and B | [110,158] |
| EMARaV | Species-specific RT–PCR, EBIA, qRT–PCR, Dot blot hybridization | [7,14,22,23,69,79,167] |
| FMV | Species-specific RT–PCR, RT–LAMP, electrochemical immunosensor, Western Blot, Dot immuno-binding and DAS–ELISA | [25,125,147,168–170] |
| RRV | ELISA, immuno dip-stick, IC–RT–PCR, multiplex RT–PCR, quantitative RT–PCR, RT–RPA, LAMP | [26,90,171–176] |
| RLBV | Species-specific RT–PCR, Dot-blot hybridization | [28,29,149] |
| PPSMV-1 | Species-specific RT–PCR, DAS–ELISA, DIBA, ELISA | [11,12,31,121,177] |
| PPSMV-2 | RT–PCR, ELISA | [31,32,121] |
| HPWMoV | Species-specific RT–PCR, ELISA (AGDIA), Northern Blot | [33,178] |
| RYRaV | Species-specific RT–PCR | [34] |
| AcCRaV | Species-specific RT–PCR, Multiplex RT–PCR | [35,179] |
| BLMaV | Species-specific RT–PCR, qRT–PCR | [36,180] |
| PiVB | Species-specific RT–PCR | [37] |
| PVBV | Species-specific RT–PCR | [38] |
| AcV-2 | Species-specific RT–PCR | [39] |
| JYMaV | Species-specific RT–PCR targeting RNA 3-RNA 6, virus protein detection with antibody by Western Blot | [40,41,153] |
| TiRSaV | RT–PCR using genus-specific primers and species-specific primers targeting RNA 1 | [42] |
| CjaV-1 and -2 | Species-specific qRT–PCR, RT–PCR targeting RNA 1, RNA 5 and RNA 6 | [43,45] |
| PerMV | Species-specific RT–PCR, immunoblot with specific antibody raised against peptide of viral P3 protein | [46] |
| AsMaV | Species-specific RT–PCR | [17] |
| LiCRaV | Species-specific RT–PCR | [47] |
| PCLSaV | Species-specific RT–PCR | [48] |
| CORaV | Species-specific RT–PCR | [18,19] |
| MaMaV | Species-specific RT–PCR | [20] |
| ChMaV | Species-specific RT–PCR | [50,110] |

**Table 2.** *Cont.*

| Species | Detection Assay | Reference(s) |
|---------|-----------------|--------------|
| ARaV | HTS-based detection | [51] |
| VEV | Species-specific RT–PCR | [53] |
| ASaV | Species-specific RT–PCR | [21] |
| KOPV | Species-specific RT–PCR | [56] |
| JSARaV | Species-specific RT–PCR | [57] |
| ArSaV1 | HTS-based detection only | [58] |
| ArtV1 | HTS-based detection only | [59] |
| ACrLaV | HTS-based detection only | [60] |
| PloAEV | Species-specific RT–PCR | [61] |

*9.2. Control Strategies*

To prevent further spread of emaraviruses, we suggest four precautions: (1) Establish a routine assay for specific diagnosis of each emaravirus, with regulations for imported plant material. These assays should be based on reliable methods such as ELISA or RT–PCR. Thanks to such assays, FMV (maybe the best studied emaravirus because of its worldwide occurrence) is being detected in an increasing number of areas [96,181]. (2) Comply with hygiene measures for emaraviruses that are mechanically transmissible. If suspected symptoms arise, affected plants should be tested and removed if an infection is confirmed. (3) Combat the mite vectors in the field, since gall mites are the main transmission factor. (4) Test innovative approaches. For PPSMV, a first study successfully made use of double-stranded RNA to protect plants against the virus infection [182].

**10. Research Requirements for Emaraviruses**

Emaraviruses came into the focus of plant virologists owing to their worldwide distribution and their ability to infect a broad range of host plants including important crops. However, we still know little about many aspects of their biology and evolution. To fill these gaps, we propose the following six main research needs:

(1)  Mapping all genome segments for each emaravirus;
(2)  Determining the functions of proteins encoded by non-core genome segments;
(3)  Understanding emaravirus–plant interactions;
(4)  Breeding resistant plants;
(5)  Clarifying transmission modes;
(6)  Investigating the evolutionary link between host range, transmission, and biology.

We discuss these research needs below.

1.  Mapping all genome segments. The bioinformatic analysis presented here (Section 6.2) strongly suggests that some "non-core" genome segments might have been overlooked in some emaraviruses. One of our first priorities should be to map them and to ensure that no genome segment is overlooked when a new emaravirus is discovered. Our description of non-core segments typically found in some clades will help to achieve this aim.
2.  Determining the function of proteins encoded by non-core segments. The development of reverse genetic tools, recently introduced for RRV, has opened new possibilities for exploring the roles of emaravirus proteins [119]. This reverse-genetics approach can also be used to determine the function(s) of proteins encoded by non-core segments. The homologies we have identified and the conserved domains shared between many accessory proteins should help to accelerate this process, by suggesting functions common to homologs.

3.  Understanding emaravirus–plant interactions. The newly developed reverse genetic tools should also help us map molecular interactions between emaraviruses and their hosts or vectors [119]. For example, Pang et al. [93] observed different symptoms between *Arabidopsis* plants experimentally infected with *Agrobacterium*-delivered RRV cDNA clones and between naturally infected roses or infected *N. benthamiana*.

4.  Breeding resistant plants. Reverse genetics will also accelerate progress in breeding research to identify resistant host genotypes. Such research is essential for emaraviruses that cause strong damage to affected plants, but was until now painstakingly slow.

5.  Clarifying transmission modes. As long as the modes of transmission of emaraviruses are not fully characterized, no comprehensive strategy can be developed to combat them. Additional factors that contribute to virus spread cannot be ruled out at that time. They would tremendously influence the way control approaches can be best implemented. In particular, determining whether emaraviruses can be transmitted through water and soil would be especially interesting for long-living woody hosts in which the role of viruses on the overall health status is harder to elucidate [18].

6.  Investigating the evolutionary link between host range, transmission, and biology.

Although more phylogenetic data is needed to reconstruct emaravirus evolution, a number of points are already clear. There is a poor correlation between the phylogeny of viruses and their hosts, suggestive of host shifts during evolution. For example, RRV, EMARaV, RLBV, and PCLSaV all infect plants in the *Rosaceae* family, but cluster in four phylogenetic clades. Likewise, the species PPSMV-1 and PPSMV-2 (or AcCRaV and AcV-2) infect the same host plant, but are only distantly related.

Similarly, the relationship between phylogeny and vector specificity appears complex: FMV and HPWMoV, that belong to clade A and C, respectively, are both transmitted by mites of the genus *Aceria*. Similarly, mites of the genus *Phyllocoptes* were demonstrated to transmit RLBV and RRV that belong to clade C and A, respectively. This might be suggestive of shifts in vector specificity during evolution, either as a consequence or a cause of host shifts.

As viruses are believed to act as predisposing factors and impede infected plants from coping with further stresses [166,183], special focus needs to be put on viral infections. Emaraviruses represent a great research field owing to their wide distribution and host range as well as their agricultural and ecological impact. Thanks to high-throughput sequencing, the number of emaraviruses and of their genomic components is constantly growing. Their complex symptomatology, genetic equipment, and evolution makes it necessary to invest more into their research. The advent of reverse genetics will make this investment worthwhile and allow us to understand all aspects of the biology of these fascinating group of viruses.

**Supplementary Materials:** The following supporting information are available online at http://www.mdpi.com/xxx/s1, Table S1: Accession numbers used for phylogenetic analysis, Table S2: Accession numbers of emaraviral proteins containing the CTR region, Table S3: Accession numbers of viral proteins containing the Glu2–Pro domain, Table S4: Accession numbers of orphan sequences.

**Author Contributions:** Conceptualization, M.R., S.v.B. and C.B.; methodology, M.R.; software, M.R. and D.G.K.; meta analysis, M.R.; validation, M.B., S.N.Z., T.C. and C.B.; formal analysis, M.R.; investigation, M.R., D.G.K. and R.A.K.; resources, S.v.B. and C.B.; data curation, D.G.K.; writing—original draft preparation, M.R.; writing—review and editing, D.G.K., M.B., R.A.K., S.N.Z., T.C., C.B. and S.v.B.; visualization, M.R., D.G.K. and S.v.B.; supervision, S.v.B.; project administration, C.B.; funding acquisition, M.R., R.A.K. and C.B. All authors have read and agreed to the published version of the manuscript.

**Funding:** This research was essentially funded by FAZIT-STIFTUNG Gemeinnützige Verlagsgesellschaft mbH (personal grant to Marius Rehanek, no grant number provided). This work has further been funded by German Research Foundation (DFG), grant numbers BU890/27-1, MU559/13-1, BU890/31-1, EINSTEIN Foundation, grant number EGP-2028-476, Agency of Renewable Resources (FNR), grant number FNR 2220WK40B4 and Hamburg Ministry for Economy and Innovation, grant number 734.650-004/014A. Research collaboration was financially supported by European Cooperation in Science and Technology (COST-DIVAS action FA1407). The article processing charge was funded by the Deutsche Forschungsgemeinschaft (DFG, German Research Foundation)—491192747 and the Open Access Publication Fund of Humboldt-Universität zu Berlin.

**Data Availability Statement:** Not applicable.

**Acknowledgments:** This review is the scientific result of intensive research on emaraviruses affecting forest tree species over 20 years, mainly at Humboldt-Universität zu Berlin. Special thanks for input and power are due to all scientists, technical assistants, and students who have accompanied our research all these years. For excellent electronic microscopic pictures, we thank Katja Richert-Pöggeler and her group at JKI (Germany), (Figures 1a and 3f) as well as Katia Gindro (Agroscope, Switzerland) (Figure 1b). We are especially grateful to Thomas Gaskin (LELF Brandenburg, Germany) for final proofreading of the manuscript. We gratefully acknowledge the financial support through funding of many projects received from the above-mentioned institutions.

**Conflicts of Interest:** The authors declare no conflict of interest. The funders had no role in the design of the study; in the collection, analyses, or interpretation of data; in the writing of the manuscript, or in the decision to publish the results.

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
