# Peer review of "The Complex World of Emaraviruses—Challenges, Insights, and Prospects"

_forests, doi:10.3390/f13111868_

Round 1
Reviewer 1 Report
The review manuscript by Rehanek et al “The complex world of emaraviruses – challenges, insights and 2 prospects” is excellently compiled, analyzed and written, however, I came across some grammatical errors, which are listed below. I also have suggested the authors if they could consider including additional relevant references.
Line No. 322: chapter 7 or section 7? Please cross check
Line No. 390: JYMaV RNA 4 and RNA 5 each encode a MP, homologous to 390 emaraviral MPs [55].
Line No. 557: 6.3. Incomplete genome information and harmonization of proteins names
Line No. 568-570: Would it be too hard to include the P8 sequence of PVBV in the sequence analysis images? If it is not going to take much effort, then you may include this also.
Line No. 628-629: Please remove the article “a” from “confers a selective advantages”.
Line No. 633: remove “and”
Line No. 708: Please also cite the reference “149”, since this publication (Patil et al., 2017) identified the presence of PPSMV-2 in several different locations across India.
You may also add the following reference: Baskar, S., Patil, B.L., Latha, T.K.S., and Karthikeyan, G. (2021) Variability studies on Pigeonpea sterility mosaic emaraviruses in Tamil Nadu reveals rampant mixed infections and interspecies recombination. Journal of Food Legumes. 34(3): 173-180.
Line No. 950-952 & 954: You may cite this publication for economic impact of plant viral diseases: Plant Viral Diseases: Economic Implications. In: Bamford, D.H. and Zuckerman, M. (eds.) Encyclopedia of Virology, 4th Edition, vol. 3, pp. 81-97. Oxford: Academic Press. http://dx.doi.org/10.1016/B978-0-12-809633-8.21307-1
In 9.2. Control strategies: You may discuss about: Patil, B.L., Raghu, R., Dangwal, M., Byregowda, M., and Voloudakis, A.E. (2021) Exogenous dsRNA-mediated field protection against Pigeonpea sterility mosaic emaravirus. Journal of Plant Biochemistry and Biotechnology. 30, 400–405. https://doi.org/10.1007/s13562-020-00627-z
For PPSMV, you may cite this reference appropriately which has a detailed description of PPSMV starting from historic times: Patil, B.L., and Kumar, P.L. (2017) Pigeonpea sterility mosaic emaraviruses: a journey from a mysterious disease to a classic emaravirus. In: A Century of Plant Virology in India. Editors: B Mandal, GP Rao, VK Baranwal & RK Jain, Springer. Chapter 10, 255-270.
At several instances, the authors have referred to the “sections” as “chapter”, I will appreciate if they could cross check this and make appropriate corrections.
Author Response
Comments reviewer 1
The review manuscript by Rehanek et al “The complex worldof emaraviruses – challenges, insights and 2 prospects” is excellently compiled, analyzed and written, however, I came across some grammatical errors, which are listed below. I also have suggested the authors if they could consider including additional relevant references.
Line No. 322: chapter 7 or section 7? Please cross check Done and harmonized throughout the manuscript (lines 84-99, line 884)
Line No. 390: JYMaV RNA 4 and RNA 5 each encode a MP, homologous to emaraviral MPs [55]. Not changed, because it has been checked by a native speaker
Line No. 557: 6.3. Incomplete genome information and harmonization of proteins names Not changed, because it has been checked by a native speaker
Line No. 568-570: Would it be too hard to include the P8 sequence of PVBV in the sequence analysis images? If it is not going to take much effort, then you may include this also.
Due to the very short time between release of this novel sequence report of PVBV P8 and the deadline of manuscript submission, we decided to just take up this “small information” into this review, the way we did, but did not want to add additional figures to this manuscript. This would have been necessary as the PVBV-P8 is an ABC protein, for which no sequence analysis figure is presented in the review, since the sequence analyses for groups A and B and group C were presented in two earlier publications; the review presents instead a novel structural analysis (Figure 10) on two representative proteins from clades A and C. Therefore, we couldn't include P8 in a sequence analysis figure.
Line No. 628-629: Please remove the article “a” from “confers aselective advantages”. Done
Line No. 633: remove “and” Done
Line No. 708: Please also cite the reference “149”, since this publication (Patil et al., 2017) identified the presence of PPSMV-2 in several different locations across India. Done
You may also add the following reference: Baskar, S., Patil, B.L.,Latha, T.K.S., and Karthikeyan, G. (2021) Variability studies on Pigeonpea sterility mosaic emaraviruses in Tamil Nadu reveals rampant mixed infections and interspecies recombination. Journal of Food Legumes. 34(3): 173-180. Not included, because the reference 149 already states the wide distribution of PPSMV-1 and PPSMV-2 in India, respectively, though this publication do not add relevant new information to the review.
Line No. 950-952 & 954: You may cite this publication for economic impact of plant viral diseases: Plant Viral Diseases: Economic Implications. In: Bamford, D.H. and Zuckerman, M.(eds.) Encyclopedia of Virology, 4th Edition, vol. 3, pp. 81-97. Oxford: Academic Press. http://dx.doi.org/10.1016/B978-0-12-809633-8.21307-1 Done
In 9.2. Control strategies: You may discuss about: Patil, B.L.,Raghu, R., Dangwal, M., Byregowda, M., and Voloudakis, A.E. (2021) Exogenous dsRNA-mediated field protection againstPigeonpea sterility mosaic emaravirus. Journal of Plant Biochemistry and Biotechnology. 30, 400–405.https://doi.org/10.1007/s13562-020-00627-z Done
For PPSMV, you may cite this reference appropriately which has a detailed description of PPSMV starting from historic times: Patil, B.L., and Kumar, P.L. (2017) Pigeonpea sterility mosaic emaraviruses: a journey from a mysterious disease to a classic emaravirus. In: A Century of Plant Virology in India. Editors: BMandal, GP Rao, VK Baranwal & RK Jain, Springer. Chapter 10,255-270. As the review has the focus on emaraviruses of woody host species we decided to describe the history of emaraviruses on the basis of EMARaV discovery.
At several instances, the authors have referred to the “sections”as “chapter”, I will appreciate if they could cross check this and make appropriate corrections. Done, see above comments
Reviewer 2 Report
A comprehensive review of the emaraviruses. I note that the authors give the new binomial species names as well as the common names. The images in the figures are clear and informative. There are a few minor grammatical issues, but other than a thorough edit, I recommend publication as it is.
Author Response
A comprehensive review of the emaraviruses. I note that the authors give the new binomial species names as well as the common names. The images in the figures are clear and informative. There are a few minor grammatical issues, but other than a thorough edit, I recommend publication as it is.
No recommendations for changes have been suggested by reviewer 2